# Developmental basis of SHH medulloblastoma heterogeneity

Maxwell P. Gold[1], Winnie Ong[2,3], Andrew M. Masteller[1], David R. Ghasemi [4,5,6], Julie Anne Galindo[7], Noel R. Park [8,9,10], Nhan C. Huynh[1], Aneesh Donde[1], Veronika Pister [1], Raul A. Saurez[2], Maria C. Vladoiu[2], Grace H. Hwang[11,12], Tanja Eisemann [13], Laura K. Donovan[2,14], Adam D. Walker[7], Joseph Benetatos [1], Christelle Dufour [15,16], Livia Garzia [17,18], Rosalind A. Segal [11,12], Robert J. Wechsler-Reya[13,19,20], Jill P. Mesirov [21], Andrey Korshunov[4,22,23,24], Kristian W. Pajtler [4,5,6], Scott L. Pomeroy[25,26,27], Olivier Ayrault [28,29], Shawn M. Davidson [8,30,31], Jennifer A. Cotter [7,32], Michael D. Taylor [2,3,14,33,34,35,36,37] & Ernest Fraenkel [1,27] ✉

Many genes that drive normal cellular development also contribute to onco-genesis. Medulloblastoma (MB) tumors likely arise from neuronal progenitors in the cerebellum, and we hypothesized that the heterogeneity observed in MBs with sonic hedgehog (SHH) activation could be due to differences in developmental pathways. To investigate this question, here we perform single-nucleus RNA sequencing on highly differentiated SHH MBs with extensively nodular histology and observed malignant cells resembling each stage of canonical granule neuron development. Through innovative computational approaches, we connect these results to published datasets and find that some established molecular subtypes of SHH MB appear arrested at different developmental stages. Additionally, using multiplexed proteomic imaging and MALDI imaging mass spectrometry, we identify distinct histological and metabolic profiles for highly differentiated tumors. Our approaches are applicable to understanding the interplay between heterogeneity and differentiation in other cancers and can provide important insights for the design of targeted therapies.

Medulloblastoma (MB) is one of the most common malignant pediatric brain tumors. The standard treatment regimen of surgical resection, radiation, and chemotherapy has led to favorable short-term outcomes in aggregate[1,2], but unfortunately these therapies can cause neurological side effects and increased risks of secondary cancers[3,4]. Thus, there is an urgent need for more targeted, less toxic therapies, which requires a better understanding of the heterogeneity within and between MB tumors[5–8].

The World Health Organization recognizes both histological and molecular heterogeneity in MB[9,10]. The four primary histological categories are classic, large cell anaplastic (LCA), desmoplastic/nodular (DNMB), and medulloblastomas with extensive nodularity (MBEN). DNMB histology is characterized by tightly packed cells interrupted by nodules filled with a lower density of differentiated neuron-like cells. Tumors with widespread nodularity are designated as MBENs.

In addition to histological heterogeneity, there are four consensus molecular subgroups recognized by the MB research community[11]: WNT, SHH, Group 3, and Group 4. SHH MBs represent 30% of cases and have an overactive sonic hedgehog pathway caused by germline or acquired mutations[12]. Granule cell precursors (GCPs) of the cerebellum are the proposed cell of origin for these tumors[13–15]. During normal development, the GCPs proliferate in response to SHH in the external

granule layer (EGL)[16,17] before differentiating into granule neurons (GNs)[18], which then migrate to their final location in the internal granule layer (IGL)[19–21].

Many groups have characterized the molecular heterogeneity of SHH MBs. Analysis of methylation and transcriptional data revealed four consensus subtypes: SHH-1 (β), SHH-2 (ɣ), SHH-3 (α), and SHH-4 (δ)[5,22]. Additionally, Archer et al. found proteomic clusters of SHH MB tumors[23] and Korshunov et al. identified a transcriptional subtype of SHH MBEN tumors with exceptionally good outcomes[24]. Single-cell RNA sequencing (scRNA-seq) studies have also characterized the cell types in SHH MBs[6–8], where they observed undifferentiated progenitors resembling cerebellar GCPs and differentiated NeuN+ cells.

Since SHH MBs are proposed to originate from GN progenitors, we hypothesized that the inter- and intra-tumoral heterogeneity in these samples is related to their developmental origins. Single-cell clusters and molecular subtypes have been described as differentiated, but not associated with specific stages of GN development. Precise annotations would allow for more biologically informed discussions about the clinical and therapeutic relevance of specific cell types. For example, there is great interest in using differentiation therapy to treat SHH MB by inducing cycling progenitors to differentiate into post-mitotic neurons[25–27], and more knowledge about the drivers of differentiation in these tumors would help inform target identification.

We reason that the relationship between development and intra-tumoral heterogeneity would be particularly pronounced in tumors with MBEN histology because they contain widespread differentiated nodules. Therefore, in this work, we perform single-nucleus RNA sequencing (snRNA-seq) on seven SHH MBs with the MBEN histology. We identify cells mimicking every stage of cerebellar GN development and then use computational techniques to relate these MBEN cell types to previously described examples of SHH MB heterogeneity[5–8,23,24]. Specifically, we detail insights about tumor subtypes, copy number variations, metabolism, and histology. Overall, this work highlights computational and experimental approaches that can be used to investigate connections between tumor heterogeneity and known developmental trajectories.

## Results

### Tumor cells in Medulloblastomas with Extensive Nodularity (MBEN) recapitulate granule neuron development

Prior scRNA-seq studies of SHH MB have not included any tumors with MBEN histology[6–8]. While rare, such tumors are of particular interest due to their occurrence in very young patients and their unusual morphology, characterized by large regions of differentiated cells. We reasoned that a deeper analysis of MBEN samples might provide insight into the relationship between normal GN development and tumor differentiation, so we performed snRNA-seq on seven MBEN tumors (Supplementary Data 1).

First, we clustered the data and annotated the corresponding cell types (Fig. 1a). Tumor cells represent 92% of the high-quality nuclei, while the remaining nuclei are from cell types that commonly infiltrate SHH MBs, such as macrophages and microglia (Fig. 1a, Supplementary Fig. 1A). The malignant nuclei are heterogeneous and Louvain clustering revealed eight groups of cells (Supplementary Fig. 1B). Our data confirm previous reports of tumor cells resembling non-cycling GCPs (GLI2 + /TOP2A-), cycling GCPs (GLI2 + /TOP2A + ), and differentiated cells (NeuN + ) (Fig. 1b). Additionally, we performed pseudotime analysis on the malignant nuclei and identified a clear trajectory from cycling GCPs to differentiated cells (Fig. 1b).

Despite these patterns, pseudotime and UMAP plots are not necessarily reflective of genuine biological differentiation. Fortunately, canonical GN differentiation is well studied in both humans and rodents, and there are known marker genes for each developmental stage (Fig. 1c). GCPs express the SHH pathway marker GLI2 and proliferate in the EGL. In the normal developing cerebellum, the actively cycling GCPs (TOP2A +) are located in the outer EGL, while the non-cycling GCPs reside in the inner EGL[16,17,28]. The GCPs can differentiate into GN, where they are SEMA6A+ for the short time they are migrating tangentially within the EGL[29]. The GN then express the GRIN2B glutamate receptor as they migrate radially across the molecular layer (ML)[30–32]. Once they reach the IGL and start to mature, GRIN2B is replaced by the GRIN2C receptor[33–36].

To assess the relationship between the tumor cells and normal GN differentiation, we re-analyzed the nuclei along the potential GCP-to-GN trajectory. Remarkably, we observed tumor nuclei expressing key markers from every stage of GN differentiation and maturation (Fig. 1d, Supplementary Fig. 2)[37–40]. These transcriptomic patterns suggest that some MB tumor cells retain the capacity to recapitulate canonical GN development.

Since there are known biases in single-cell and single-nucleus sequencing[41], we also performed scRNA-seq on fresh cells from one MBEN tumor (Supplementary Fig. 3A). The scRNA-seq cells do not express the same exact markers as the snRNA-seq nuclei, but we still observe clusters of malignant cells with markers for each stage of GN development (Supplementary Fig. 3). Additionally, we observed a difference in cell type proportions between the two assays. For the five snRNA-seq samples that contain all stages of GN development, 66% of the tumor cells resemble migrating and postmigratory GNs. In the one scRNA-seq sample, these late-stage GNs only represent 33% of the malignant cells. This difference may be due to sample-to-sample variability or technical differences between cells captured by each assay[42,43].

### Clustering of gene set signatures reveals connections between granule neuron development and SHH MB heterogeneity

Since we observed tumor cells expressing markers from each stage of GN development, we sought to understand how these MBEN cell types relate to previously published examples of SHH MB heterogeneity[5,6,8,23,24,44]. To accomplish this, we developed the computational approach outlined in Fig. 2a. This method is based on generating gene signatures from relevant developmental and tumor datasets and then using a large compendium of expression data from SHH MBs to identify which signatures are activated in the same patients.

We considered signatures from six studies related to SHH MB or GN development[5,6,8,23,24,44] (Supplementary Fig. 4). Cavalli et al.[5] defined four consensus subtypes from bulk RNA and methylation; SHH-3 (α) and SHH-4 (δ) are associated with older patients, while SHH-1 (β) and SHH-2 (ɣ) tumors are often more neuronal and come from younger patients. Archer et al.[23] identified proteomic subtypes where SHHa tumors have proliferation and ribosomal markers, while SHHb samples have elevated levels of proteins related to synapses and glutamate signaling. Korshunov et al.[24] uncovered the TCL1 and TCL2 transcriptional subtypes of SHH MBEN samples, where TCL2 tumors have high expression of neuronal genes and come from patients with exceptional survival rates.

In addition to these three bulk omics experiments, we also included cell types from three scRNA-seq studies. Riemondy et al.[8] analyzed SHH MBs and found six clusters of tumor cells. SHH-A cells are associated with proliferation (SHH-A1 with S phase and SHH-A2 with G2M phase), while SHH-B cells are progenitors, with SHH-B1 resembling GCPs and SHH-B2 having high ribosomal expression. Lastly, SHH-C1 cells are enriched for RNA processing and axo-dendritic transport genes, while SHH-C2 cells have high levels of neural development markers like STMN2. We also generated signatures for cell types found during cerebellar GN development and considered studies from both human[44] and mouse[6].

For each cell type or molecular subtype, we generated a signature of 100 marker genes (see Methods) (Supplementary Data 2). We then used Gene Set Variation Analysis (GSVA)[45] to calculate an activation

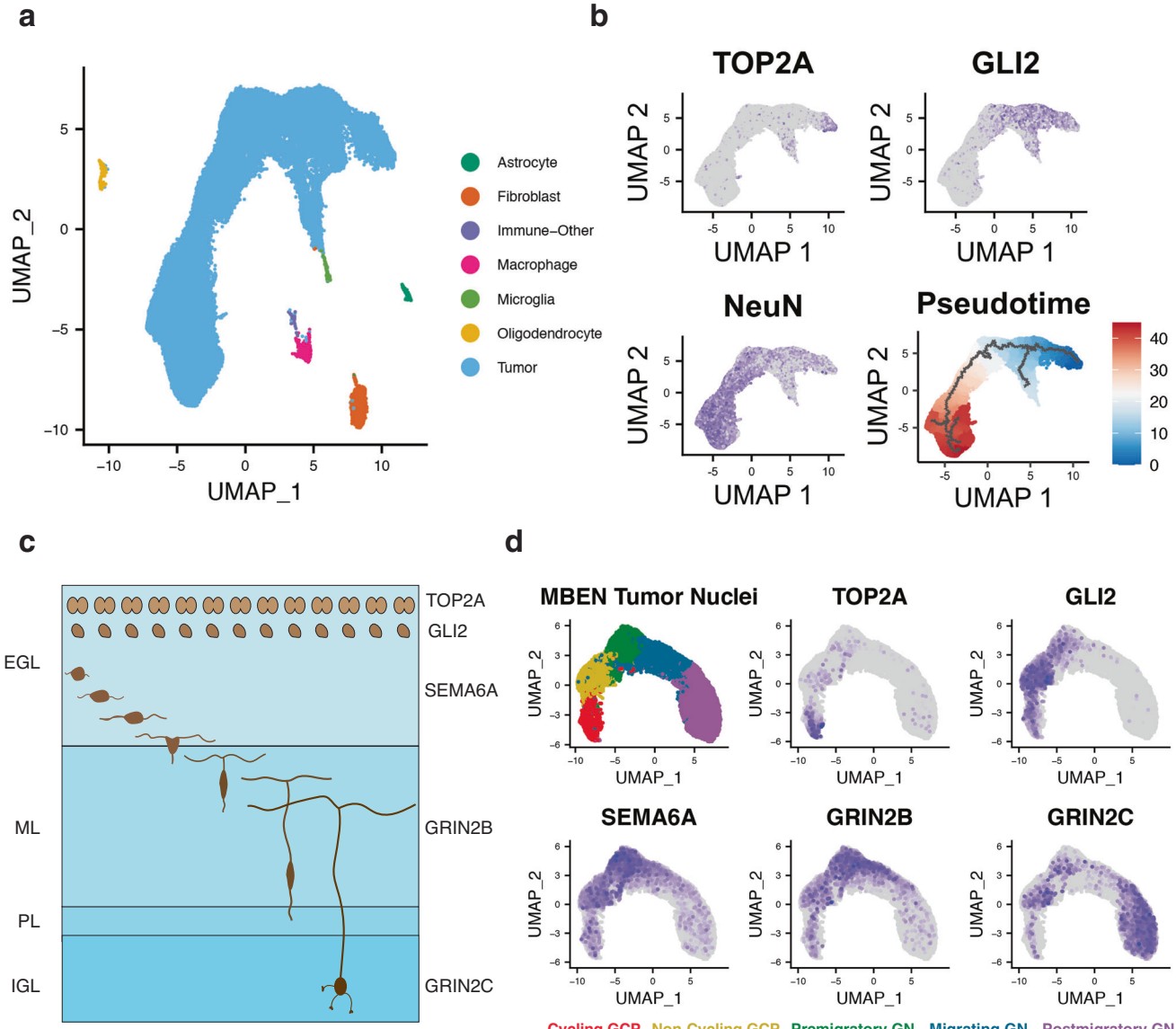

**Fig. 1 | MBEN tumor cells mimic granule neuron development. a** *Summary UMAP of Seven Tumors with MBEN Histology.* Malignant and non-malignant cell types are labeled in the legend. **b** *Marker Genes and Pseudotime for Malignant Cells.* Tumor cells express markers for cycling GCPs (TOP2A + /GLI2 +), non-cycling progenitors (TOP2A-/GLI2 +), and differentiated neurons (NeuN +). Feature plots use blue to highlight all cells above 5th percentile of expression. Bottom right image shows pseudotime analysis rooted at cycling cells. Pseudotime increases from dark blue to white and then dark red. The black lines represent trajectories identified by *monocle3*. **c** *Summary of Canonical Granule Neuron Development.* Image adapted from Consales et al.[54]. Granule cell precursors (GCPs) proliferate in the outer portion of the external granule layer (EGL), while non-cycling progenitors lie in the middle portion of the EGL. As the granule neurons (GN) differentiate, they express SEMA6A as they migrate tangentially across the inner portion of the EGL. The GN then turn and migrate radially across the molecular layer (ML), during which they express the glutamate receptor GRIN2B. Once the GN reach their final location in the internal granule layer (IGL), GRIN2C replaces GRIN2B. **d** *MBEN Tumor Cells Resemble Stages of Granule Neuron Development.* UMAP plot for malignant tumor cells along potential GCP to GN trajectory. There are tumor cells that express markers for each stage of GN development: cycling GCPs (TOP2A +), non-cycling GCPs (TOP2A-/GLI2 +), premigratory GN (SEMA6A +), migrating GN (GRIN2B +), and postmigratory GN (GRIN2C +). Feature plots use a minimum cutoff at the 90th percentile for each marker to highlight the cells with the highest expression.

score for each signature in all 223 SHH MBs from the MAGIC cohort[5] (Supplementary Data 3). To identify relationships among the signatures, we performed consensus clustering on the signature scores. Figure 2b summarizes the co-clustering pattern for 1000 trials. We observe a clear correspondence between the MBEN cell types and the known stages of GN development and this pattern holds true across many clustering parameters and gene signature sizes (see Methods, Supplementary Fig. 6).

Additionally, we analyzed how the signatures of GN development relate to cell types identified from SHH MBs with non-MBEN histology. Specifically, we investigated the SHH-A, SHH-B, and SHH-C clusters from Riemondy et al.[8]. We confirmed their findings that the SHH-A and SHH-B signatures correspond to cycling and non-cycling GCPs respectively. We also identified an association between the SHH-C2 signature and premigratory GN, which is supported by previous studies showing that the SHH-C2 cells populate the differentiated nodules in SHH MBs[8].

## Consensus subtypes of SHH MB are associated with specific developmental stages

The signatures we identified from normal GN development and MBEN tumors provide an opportunity to connect these cell types to other examples of SHH MB heterogeneity. Clustering of DNA methylation and transcriptomics data revealed four consensus subtypes of SHH MB

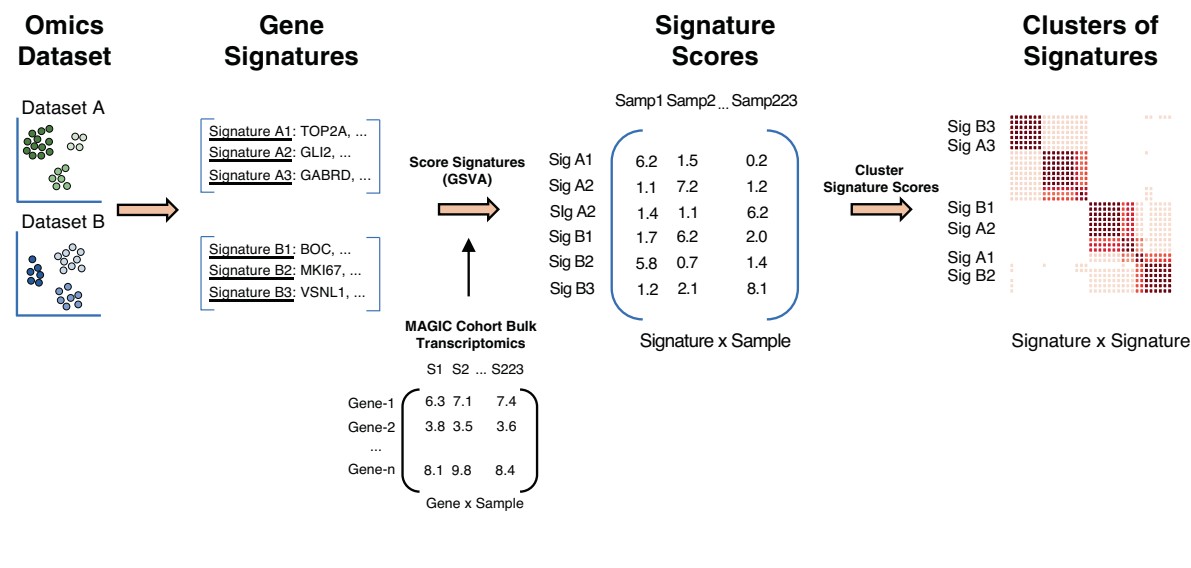

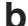

with distinct clinical and molecular features: SHH-1 (β), SHH-2 (ɣ), SHH-3 (α), and SHH-4 (δ)[5]. The 223 SHH tumors from the MAGIC cohort are annotated with a SHH subtype, so we re-analyzed the GSVA activation scores for our MBEN cell types to investigate potential associations (Fig. 3a). The SHH-3 samples have the highest average cycling GCP signature scores (0.34 in SHH-3 vs. −0.11 in others), while the SHH-4 tumors are associated with the GCP signature (0.26 vs. −0.10). Both

infant subtypes (SHH-1 and SHH-2) have high scores for differentiated cells but are enriched for specific developmental stages; SHH-1 samples have the highest premigratory GN scores (0.43 vs. −0.08), while SHH-2 tumors have significant upregulation of the postmigratory GN signature (0.50 vs. −0.21). It is noteworthy that the subtype with the least favorable outcomes (SHH-3) also has the highest cycling GCP scores (Fig. 3a).

**Fig. 2 | Clustering of gene signature activation scores. a** *Clustering of Gene Signature Scores Approach.* The first step in this method is collecting the bulk or single-cell datasets of interest (Supplementary Fig. 4). For each dataset, clustering was used to identify relevant cell types or molecular subtypes. Then the top 100 marker genes were identified for each cluster to generate gene signatures. Those marker genes were used as gene sets for Gene Set Variation Analysis (GSVA), which was run on bulk transcriptomics data from 223 SHH tumors in the MAGIC cohort. This method converts the genes by samples matrix into a signatures by samples matrix by scoring each gene set signature for each of the 223 tumors. Consensus clustering was then run on those signature scores to identify which signatures are activated in

the same SHH tumors. **b** *Signature Score Consensus Clustering Summary.* Consensus clustering results of 1000 trials using the signatures by samples output dataset. For each individual clustering, 50% of the SHH MBs were randomly chosen. Beige signifies signatures that never cluster together, and stronger red coloring indicates signatures that cluster together more often. The legend at the bottom indicates the dataset of origin, the species of the cells (human or mouse), the data type (scRNA-seq, snRNA-seq, or bulk), and material type (SHH MB tumor or healthy cerebellum). The groups of signatures on the right were manually annotated using the known cerebellar cell types from human and P14 mouse samples.

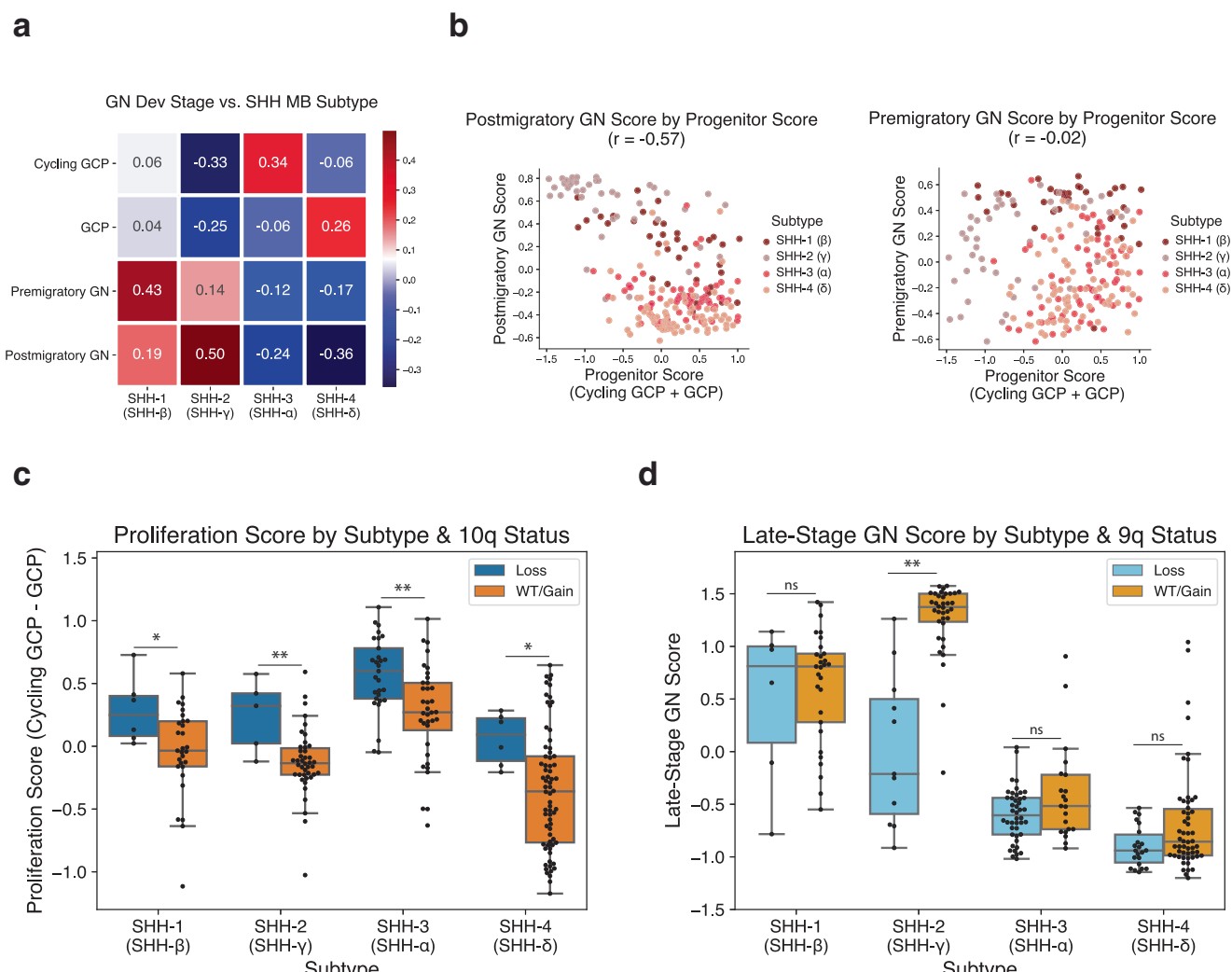

**Fig. 3 | Genomic and transcriptomic associations between SHH MB and GN development. a** *Mean Cell Type Activation Per Tumor Subtype.* Each box shows the mean GSVA activation for a given consensus subtype and an MBEN cell type. Red indicates a higher GSVA score, while blue signals a lower one. **b** *Associations Between Progenitor Score and Differentiated Cell Types.* For both plots, each dot indicates a single sample from the MAGIC cohort and is colored by consensus subtype. The left figure shows the postmigratory GN score on the y axis and the progenitor score (cycling GCP + GCP) on the x axis. These two features have a significant negative correlation. The right plot uses the same x axis, but the y axis indicates the premigratory GN score and these two variables are not significantly correlated. **c** *Boxplot of Association Between Chromosome 10q and Proliferation Score.* Boxplots reflect GSVA scores for SHH MB tumors from MAGIC cohort. Y axis is the proliferation score (cycling GCP - GCP). The x axis is separated by consensus subtype and further divided by 10q status (loss in blue, WT or gain in orange). For

all subtypes, samples with 10q loss show higher average proliferation score than the other tumors. The respective *p* values for two-sided Mann Whitney U tests from left to right are 0.027, 0.0005, 0.009, and 0.026. * and ** indicate two-sided Mann-Whitney *p* values less than 0.05 and 0.01 respectively. The boxplots from left to right show the following number of samples: 6, 29, 5, 42, 29, 36, 6, and 70. Each boxplot displays data quartiles, excluding outliers beyond 1.5 x IQR. **d** *Boxplot of Association Between Chromosome 9q and Late-Stage GNs.* Y axis is the late-stage GN score (migrating GN + postmigratory GN). The x axis is separated by consensus subtype and further divided by 9q status (loss in light blue, WT or gain in light orange). SHH-2 samples show a substantial difference in postmigratory GN signature based on 9q status. ** indicates two-sided Mann-Whitney *p* value < 0.01 (*p* = 6.5E-6). The boxplots from left to right show the following number of samples: 6, 29, 11, 36, 44, 21, 22, 54. Each boxplot displays data quartiles, excluding outliers beyond 1.5 x IQR.

We then analyzed the correlations among the MBEN cell type signatures and found that the postmigratory GN signature has a strong negative correlation ($r = -0.57$) with the progenitor score (i.e., sum of cycling GCP score and GCP score) (Fig. 3b). This suggests that tumors with more postmigratory GNs may have fewer progenitor cells. However, this pattern does not exist for the premigratory GN signature, which has no significant relationship with the progenitor score GN ($r = -0.02$) (Fig. 3b).

## Genomic associations with specific developmental stages

SHH MB tumors frequently contain copy number variations (CNVs), where large parts of chromosomal arms are lost or gained[46]. Since these chromosomal alterations are quite common and affect many genes at once, we hypothesized some CNVs could affect tumor cell differentiation. We investigated potential associations between large chromosomal CNVs and the MBEN cell types using data from the MAGIC cohort[5]. We considered individual cell type signatures and aggregate features that combine related developmental stages[5,6,8,23,24,44] (see Methods). The strongest association is between a loss of chromosome 10q and the proliferation score (cycling GCP score minus the GCP score) (Mann Whitney U test $p < 0.01$) (Fig. 3c, Supplementary Data 4). This pattern of 10q loss leading to higher cycling GCP scores is consistent for every subtype, suggesting that loss of chromosome 10q, which contains the *SUFU* and *PTEN* genes, may drive progenitor cells to remain proliferative.

Moreover, there is a strong negative relationship between a loss of chromosome 9q, which contains the *PTCH1* and *ELP1* genes, and the signature for late-stage GNs (i.e. sum of migrating and postmigratory GN scores) (Mann Whitney U test $p < 0.01$) (Fig. 3d). This pattern is most prominent in the SHH-2 patients, where the samples with high activation scores for late-stage GNs rarely have loss of chromosome 9q. This trend is not observed for other common CNVs or premigratory GNs (Supplementary Fig. 7), suggesting that loss of chromosome 9q may inhibit tumor cells from progressing to the later stages of GN differentiation.

## The SHHb proteomic subtype is associated with tumor cells mimicking late-stage granule neurons

Archer et al. identified proteomic subtypes of SHH MB that are not observed when clustering RNA or methylation data[23]. The SHHb subtype is characterized by proteins related to specific neuronal functions like glutamatergic synapses and axon guidance. We re-analyzed the Archer proteomics data and found that even though synaptic proteins, like PCLO and DLG4, are upregulated in SHHb tumors, many markers of neuronal differentiation, such as NEUROD1[47] and SEMA6A[29], show similar expression across the groups (Fig. 4a). This suggests that the SHHb proteomic subtype is not simply a proxy for tumors with differentiated cells.

To better understand the relationship between the snRNA-seq results and the SHHb phenotype, we used the SHHb proteomic markers (Supplementary Data 2) to calculate an activation score for each MBEN nucleus (see Methods). The highest activation occurs in tumor cells mimicking the migrating and postmigratory stages of granule neuron development (Fig. 4b). We then analyzed the cell type composition of each tumor individually and observed a striking pattern: most MBEN tumors contain all GN cell types, but two samples only contain nuclei resembling GCPs and premigratory GN (Fig. 4c).

These results suggest that the SHHb subtype is driven by the presence of cells resembling the latest stages of GN differentiation. This would explain why early differentiation markers have similar expression between SHHa and SHHb tumors, while proteins related to late-stage developmental processes (e.g. synaptogenesis) are significantly enriched in SHHb samples. To further investigate this trend, we performed additional snRNA-seq on six SHH MBs tumor that have known proteomic subtypes and do not have MBEN histology. In this

cohort, the three SHHa tumors (MB002, MB009, and MB019) are primarily composed of GLI2+ cells and SEMA6A+ cells resembling earlier stages of GN development, whereas all three SHHb tumors (MB005, MB015, and MB084) contain a large proportion of GABRD+ cells that resemble late-stage GNs (Supplementary Fig. 8).

We then sought to investigate how common these late-stage GN cells are in non-MBEN tumors using previously published scRNA-seq data from 14 SHH MBs[6–8]. In this cohort, no sample contains a cluster with high expression of the postmigratory GN markers GABRD and VSNL1 (Supplementary Figs. 9–14). This lack of expression is unlikely to be due to the differences between scRNA-seq and snRNA-seq because the one MBEN tumor in our cohort with scRNA-seq data clearly shows a group of GABRD +/VSNL1+ cells (Supplementary Fig. 15). It is noteworthy that even though the published non-MBEN tumors do not contain cells mimicking late-stage GNs, every sample has a cluster of SFRP1+ cells resembling undifferentiated progenitors. Additionally, many tumors have cells expressing the premigratory GN markers STMN2 and SEMA6A. These findings suggest that non-MBEN SHH MBs still contain cells resembling the earliest stages of GN development, but MBEN tumors are more likely to have cells mimicking late-stage GNs that are associated with the SHHb subtype.

## FMRP-induced post-transcriptional regulation helps explain SHHb proteomic phenotype

The association between the proteomic SHHb subtype and late-stage GNs raises another question: why does the presence of late-stage GNs result in clear clustering in the proteomic data, while having much less of an impact on RNA or methylation data? We hypothesized that this could be due to post-transcriptional regulation occurring in the SHHb-specific cell types. To test this theory, we re-analyzed 8674 genes from the Archer cohort by rank-normalizing the protein and RNA data for each sample and then calculating a rank difference (protein rank – RNA rank) for each gene to get a rough proxy for post-transcriptional regulation (see Methods).

We explored many gene sets and found that synaptic genes have especially high rank differences in the SHHb tumors, but not the SHHa tumors (Supplementary Fig. 16). We then specifically analyzed targets of FMRP, a protein encoded by the *FMR1* gene that regulates the nuclear export and translation of RNAs related to neuron development, synaptic plasticity, and axon guidance[48–50] (Supplementary Fig. 17). In SHHb tumors, synaptic genes targeted by FMRP have significantly higher rank differences than all other genes and even show higher rank differences than the other synaptic genes (Fig. 4d). No such trend occurs in SHHa tumors where all four gene sets show similar rank differences around 0. Thus, positive rank differences for both synaptic genes and FMRP targets suggest that the SHHb phenotype may be especially strong in proteomic data due to post-transcriptional regulation specific to functions of late-stage GNs.

## Desmoplastic/Nodular (DNMB) histology in SHH MB reflects granule neuron development

We sought to understand how the neuronal MBEN cells from our snRNA-seq data relate to the differentiated nodular regions observed in some SHH MBs. Eberhart et al. compared DNMB histology to the layers of the developing cerebellum (Fig. 1c)[51], hypothesizing that the cycling internodular regions represent the progenitor cells of the EGL and that the nodules themselves represent the differentiated GN of the IGL. This model has been a useful framework for understanding DNMB histology, but our snRNA-seq data suggests it can be improved since our cohort includes two MBEN tumors that contain no cells resembling the postmigratory GN that populate the IGL (Fig. 3c).

Thus, we propose an alternative model where most SHH MB tumors with DNMB histology are composed of nodules containing NeuN+ cells mimicking premigratory GN; these regions most closely correspond to the most internal part of the EGL, rather than the IGL

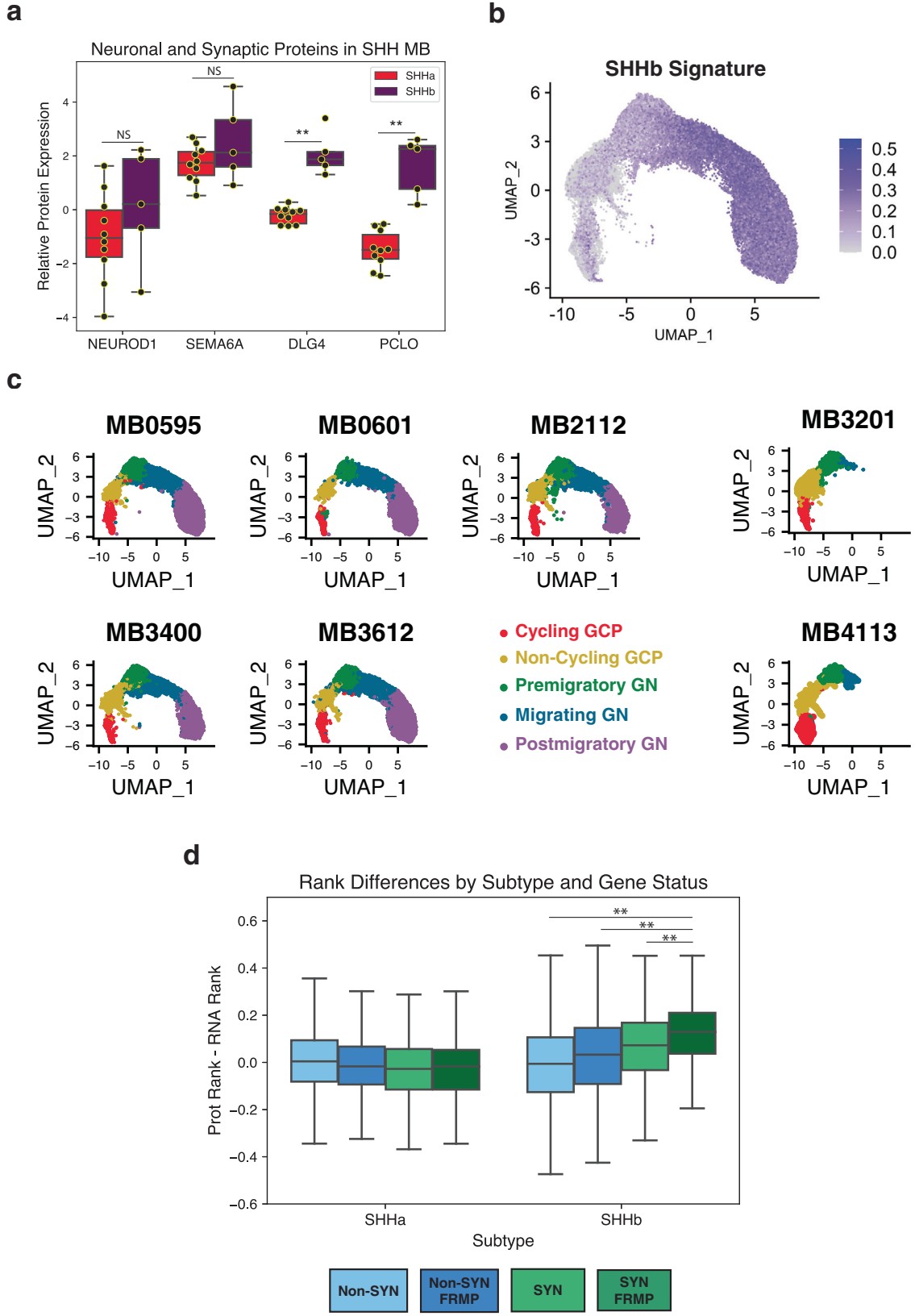

(Fig. 1c). Additionally, we posit that a subset of SHH MBs contain nodules that recapitulate the later migrating and postmigratory stages of GN development.

To test this hypothesis, we performed multiplexed immunohistochemistry (mIHC)[52,53] on SHH MBs to detect relative protein levels for four markers related to GN development: Ki67 (cycling cells), MAP2

(all GNs), CNTN1 (late-stage GN axons), and VSNL1 (late-stage GN axons and dendrites). These proteins allow us to distinguish premigratory GNs (MAP2 + /VSNL1-) from late-stage GNs (MAP2 + /VSNL1 +) (Supplementary Fig. 18). We ran this experiment on sections from eight tumors with known SHHa/SHHb proteomic subtypes[23]. All nodular tumors contain cells resembling premigratory GN (MAP2 + /VSNL1-),

**Fig. 4 | SHHb proteomic subtype associated with late-stage GNs. a** *Neuronal Protein Expression in SHH MB*. Relative protein expression from Archer et al.[23] data for markers of early differentiation (NEUROD1 and SEMA6A) and synapses (DLG4 and PCLO). DLG4 and PCLO are significantly different between SHHa and SHHb (two-sided t-test *p* values of 1.8E-6 and 1.1E-5). NEUROD1 and SEMA6A are not significantly different between subtypes. Each paired boxplot represents 10 SHHa tumors (red) and 5 SHHb tumors (purple) and shows data quartiles, excluding outliers beyond 1.5 x IQR. **b** *SHHb Signature Scores for MBEN Nuclei*. Activation scores for each MBEN nucleus using the top 100 SHHb marker proteins. Scores were filtered using a minimum cutoff of zero to highlight nuclei with the highest SHHb activation, which are the nuclei resembling migrating and postmigratory GN. **c** *Tumor Cell Types per Sample*. Five MBEN samples contain all types of cells. Two MBEN tumors do not contain late-stage GNs (migrating and postmigratory GN). **d** *Rank Differences by Synapse and FMRP status*. For each SHH tumor from Archer

et al.[23], RNA and protein data were rank normalized and a rank difference (protein rank - RNA rank) was calculated for each gene. For each gene in every SHHa tumor, the mean rank difference was calculated across samples and then the same procedure was applied to SHHb tumors. The genes were then divided into four categories: non-synaptic genes not targeted by FMRP (Non-SYN, 7392 genes), non-synaptic genes targeted by FMRP (Non-SYN FMRP, 454 genes), synaptic genes not targeted by FMRP (SYN, 628 genes) and synaptic genes targeted by FMRP (SYN FMRP, 200 genes). For SHHb tumors, SYN and SYN FMRP genes have high rank differences. For SHHa tumors, all categories show rank differences around zero. ** indicates that SYN FMRP genes have significantly different rank differences than the other gene groups. Two-sided Mann Whitney U test *p* values are 9.1E-7, 6.6E-13, and 8.7E-29 when comparing the SYN FMRP group to the SYN, NON-SYN FMRP, and Non-SYN groups respectively. Each boxplot shows data quartiles, excluding outliers beyond 1.5 x IQR.

while only the SHHb samples have areas resembling late-stage GNs (MAP2 + /VSNL1 +) (Fig. 5a). Three of the four SHHb tumors have large VSNL1+ regions, while MB206 only contains a small area of VSNL1+ cells. These findings support our proposed model and highlight that MB nodules vary in their developmental stage.

## Significant variability in VSNL1 staining between and within tumors

Korshunov et al. found that the TCL2 transcriptional subtype of MBEN tumors contains diffuse VSNL1 staining and has exceptional outcomes[24]. Based on our snRNA-seq data, VSNL1 is only expressed in tumor cells mimicking migrating and postmigratory GNs (Supplementary Fig. 18). This suggests that the TCL2 subtype may be identifying samples that are primarily composed of late-stage GNs. Given the potential clinical relevance of VSNL1 staining, we wanted to better understand the variability of VSNL1+ cells in SHH MBs. We performed immunohistochemistry targeting VSNL1 on FFPE slides from an additional seven MB with DNMB or MBEN histology and observed significant heterogeneity between and within tumors (Supplementary Fig. 19). Samples like CHLA-3 contain no VSNL1, while others like CHLA-10 are almost entirely composed of VSNL1+ regions.

We further investigated this heterogeneity by running our mIHC panel on CHLA-5, which showed significant regional variability in VSNL1 staining between sections (Fig. 5b). One region is almost entirely composed of MAP2 + /VSNL1- nodules, while another area contains mostly MAP2 + /VSNL1+ cells. This suggests that local microenvironment may affect the differentiation stage in distinct regions of the same tumor.

## Tumor cell spatial organization can recapitulate the developing cerebellum

Since we observed tumor cells mimicking the expression patterns of differentiating GNs, we also wanted to investigate whether these cells spatially organize like the developing cerebellum. First, we confirmed the established phenomenon that Ki67+ cells are primarily located in internodular areas (Fig. 5b). In multiple samples, we noticed that this pattern is particularly extreme for the VSNL1+ nodules (Supplementary Fig. 20). For example, CHLA-5 and MB287 both show significant variability in the Ki67 positivity rate between VSNL1+ and VSNL1- regions (Fisher's exact *p* value < 0.01). For CHLA-5, only 1.5% of the cells in the VSNL1+ areas are Ki67 + , compared with 23.2% of the cells in the VSNL1- areas. The same trend is observed in MB287, where the Ki67+ positivity rate is 0.3% in VSNL1+ areas and 8.2% in VSNL1- areas.

Additionally, in CHLA-10, there are regions of tumor cells that appear to organize like the developing cerebellum (Fig. 5c, d). The most external layer resembles the EGL; it contains Ki67+ cells at the outer edge and markers for premigratory GNs (MAP2 + /VSNL1-) at the inner edge. Within that area is a region analogous to the molecular layer, as it contains almost no cells and many CNTN1 + /VNSL1+ axonal

processes. Lastly, the central region corresponds to the IGL and contains VSNL1+ cells mimicking postmigratory GNs. It is noteworthy that this pseudo-cerebellar structure lacks a Purkinje cell layer as Purkinje neurons cannot differentiate from GCP stem cells.

We also investigated mIHC results for cellular patterns known to occur in MBEN tumors, such as parallel rows of nuclei surrounded by neuropil. These smaller structures also resemble the developing cerebellum by having central VSNL1+ cells surrounded by their VSNL1 + / CNTN1+ axons (Supplementary Fig. 21). In summary, these findings suggest that both simple and complex nodular structures routinely observed in SHH MB can be explained by malignant cells recapitulating specific stages GN development.

## Tumors with late-stage granule neurons have distinct metabolic profiles

We are particularly interested in SHH MB metabolism because many metabolites, like glutamate, are drivers of GN migration and differentiation[54,55]. Genomic and transcriptomic heterogeneity of SHH MBs has been well studied, but there are very few published papers analyzing SHH MB metabolism and these projects typically focus on tumor type or subtype classification using bulk metabolomics[56,57].

We used MALDI imaging mass spectrometry (MALDI-IMS)[58] to collect spatial metabolomics data on ten tumor sections collected from nine individual SHH MB patients (Supplementary Fig. 22). Seven of the nine patients were included in the snRNA-seq cohort (Fig. 1), while the other two samples are from tumors with DNMB histology. From the snRNA-seq data, we know that five samples contain tumor cells resembling late-stage GNs and we sought to understand the metabolic differences between these tumors and the others. First, we calculated the mean metabolite levels for each section and performed differential analysis. The most upregulated metabolite in tumors with late-stage GNs is N-acetylaspartic acid (NAA), which is synthesized by neurons and has high concentrations in the brain[59,60]. The most downregulated metabolites are related to nucleic acids, like UDP and adenine.

We then implemented joint graphical lasso[61] (see Methods) to generate networks of conditionally dependent metabolites in each tumor to better understand how metabolite co-expression patterns change across samples. Some core metabolite relationships, like glutamate-glutamine, were observed in the networks from every sample. We used betweenness centrality, a metric of a node's influence within a network, to identify metabolites that are especially important for each tumor. When comparing the networks from samples with late-stage GNs to the other networks, taurine has the largest mean centrality difference. Figure 6a shows select metabolite edges and highlights relationships with taurine that are overrepresented in the tumors with late-stage GNs. The prominence of taurine in these tumor networks is noteworthy because of prior literature linking the molecule to GCP differentiation and GN migration[62,63]. One study specifically shows that

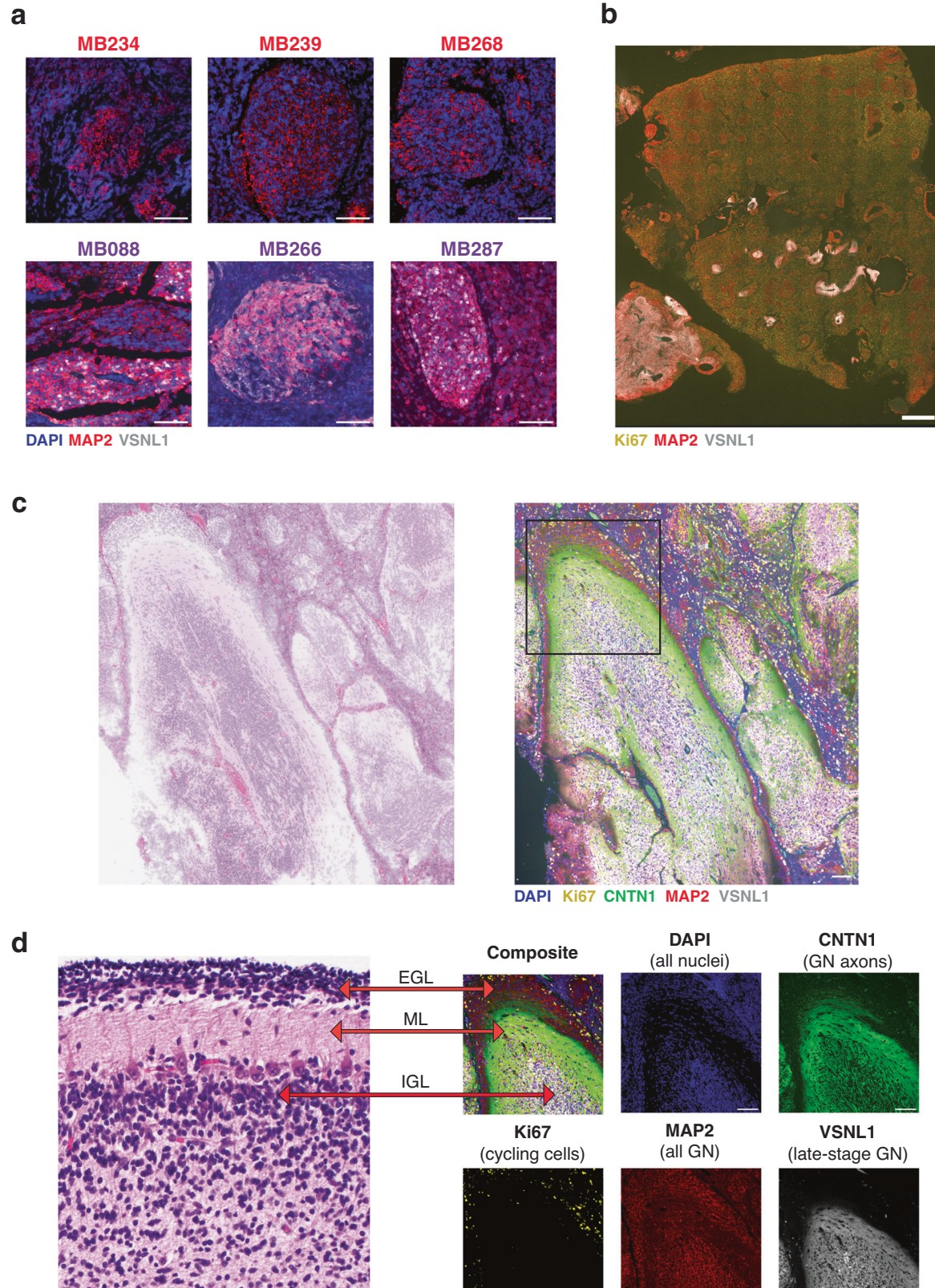

depleting dietary taurine in mother cats impedes GN development in newborn kittens, a phenotype that can be abrogated by directly feeding taurine to the kittens[63].

Given the significance of taurine in our network analysis and canonical GN development, we further analyzed the spatial distribution of taurine. We used the bivariate Moran's I statistic to assess what metabolites correlate or anticorrelate with taurine expression in a given cellular neighborhood (see Methods). We found that there is a strong negative association between taurine and guanine in tumors with late-stage GNs (Fig. 6b). This anticorrelation is visually striking in the tumors with late-stage GNs, but is not observed in other samples (Fig. 6c).

**Fig. 5 | SHH MB nodules recapitulate granule neuron development. a** *Examples of VSNL1- and VSNL1+ Nodules*. Staining for DAPI (blue), MAP2 (red), and VSNL1 (white). Each image shows a nodular region from an individual sample. The top row shows nodules with MAP2 + /VSNL1- cells resembling premigratory GN in tumors with SHHa proteomic subtype. The three tumors with the SHHb proteomic subtype are on the bottom row and contain differentiated regions that are MAP2 + /VSNL1+ mimicking the later stages of GN development. Scale bars indicate 100 μm. **b** *Ki67 and VSNL1 Anticorrelate in CHLA-5*. mIHC staining for tissue section from one sample (CHLA-5). Ki67 (yellow), MAP2 (red) and VSNL1 (white). Tissue section on bottom left is primarily composed of MAP2 + /VSNL1+ cells and has very few cycling cells. The larger tissue on the right contains many Ki67+ cells and MAP2 + /VSNL1-

nodules. Scale bars indicate 1 mm. **c** *Tumor Cells Mimic Cerebellar Structure in CHLA-10*. mIHC staining for one tissue region from CHLA-10 tumor. Left image contains H&E stain from pseudo-cerebellar region in CHLA-10. Right image shows mIHC staining same region. Scale bars indicate 100 μm. **d** *Zoomed in Region Highlights Cerebellar Layers*. Left image is H&E stain from a healthy developing cerebellum. The right images contain the boxed section from Fig. 5c, which highlights one tumor region from sample CHLA-10. Outer layer resembles the EGL and contains Ki67+ cycling cells and MAP2 + /VSNL1- cells mimicking premigratory GN. The next layer is like the molecular layer (ML), which has few nuclei and CNTN1 + /VSNL1+ axons. The white interior region represents the internal granule layer (IGL) and is filled with VSNL1+ cells mimicking postmigratory GN. Scale bars indicate 100 μm.

These results suggest that taurine may be playing a specific role in tumors with late-stage GNs. We assessed this hypothesis by performing immunohistochemistry on eleven pediatric medulloblastoma tumors. In each case, we measured fluorescence levels for MAP2, VSNL1, and taurine, and found four samples that contain both MAP2 + /VSNL1- and MAP2 + /VSNL1+ regions for comparison. Within each tumor, we observed that taurine intensities were significantly higher in the more differentiated MAP2 + /VSNL1+ regions compared to the MAP2 + / VSNL1- areas (t-test *p* value < 0.01) (Fig. 6d, Supplementary Fig. 24).

Additionally, we re-stained the pseudo-cerebellar structure from CHLA-10 with another mIHC panel including taurine and observed high taurine staining in the central region mimicking the IGL and around the edges of the structure (Supplementary Fig. 25). There appears to be a region with strong taurine staining between the pseudo-ML and the pseudo-EGL (Fig. 6e). Collectively, these imaging experiments further support that taurine is associated with tumor cells mimicking late-stage GNs.

## Discussion

We performed snRNA-seq on seven MBEN tumors and found malignant cell types mimicking every stage of cerebellar granule neuron development. By re-analyzing public MB data with this new knowledge, we were able to elucidate connections between GN development and established molecular and histological phenomena (Fig. 7). Specifically, we found that each consensus subtype of SHH MB is enriched for a specific developmental stage and that the proteomic SHHb subtype is likely caused by the presence of tumor cells resembling late-stage GNs. Additionally, a spectrum of recognized histological patterns, such as layered nodules and linear arrays of tumor nuclei in MBEN tumors, can now be understood as tumor cells mimicking the structure of the IGL of the developing cerebellum.

This work describes tumor cells mimicking late-stage GNs and presents significant progress in our understanding of the biological causes of tumor differentiation in SHH MBs. While preparing our manuscript, we became aware of a parallel study carried out by Ghasemi et al. (2023). Strikingly, they identify the same cell types that we do and observe similar spatial patterns. Additionally, Ghasemi et al. collected snRNA-seq data from three tumors from the Archer et al. proteomics cohort[23] and consistent with our predictions, these SHHb tumors (MB088, MB266, and MB287) all contain cells resembling late-stage GNs.

It is still not well understood why some SHH MBs are primarily composed of differentiated cells and other tumors have none. Based on this study, we believe that tumor microenvironment and genomics are important factors. Tumors with extensive nodularity rarely contain large CNVs and frequently occur in infants, whose brains are actively developing. We hypothesize the tumor microenvironment in these young patients may contain pro-development factors that can induce some malignant cells to escape the progenitor state and follow GN differentiation. If these tumor cells truly recapitulate GN development, they could promote the differentiation of nearby malignant cells through the release or expression of factors like glutamate and CNTN1[30,64–66]. This could potentially induce a feed-forward loop

whereby more maturation leads to more cells producing pro-differentiation molecules.

We also observed that CNVs are associated with distinct developmental stages. We highlighted alterations to chromosome 9q and 10q, which are significant because they contain the key SHH pathway genes *PTCH1* and *SUFU*. Loss of 9q or 10q can activate the SHH pathway, but these CNVs may also promote tumorigenesis by inhibiting differentiation. *PTEN* and *NEURL1* are both located on chromosome 10q and negatively regulate GCP cycling[67–69]. Additionally, chromosome 9q contains *NTRK2*, which plays a vital role in granule cell migration[70,71] and maturation[38]. There are many SNPs and CNVs with unknown effects on SHH MBs and it is worth further exploration to determine the potential developmental impact of those mutations.

Our work reinforces the importance of understanding differentiation state for prognosis. Korshunov et al.[24] showed that MBENs with diffuse VSNL1 staining have excellent outcomes, while the other MBEN tumors have similar survival rates to samples with DNMB histology[24]. Our snRNA-seq data demonstrates that VSNL1 is exclusively expressed in cells resembling late-stage GNs, suggesting these cell types may have greater clinical relevance than other differentiated cells. This hypothesis is supported by our GSVA analysis, which shows that the late-stage GN score negatively correlates with the progenitor score; by contrast, the score for premigratory GNs, an earlier stage of differentiation, has no relationship with the progenitor score (Fig. 3b). Together, these results suggest that not all SHH MB differentiation has the same prognostic relevance, and that differentiation stage could potentially be used to stratify individual patient risk more accurately.

These molecular and prognostic insights are important for the development of therapeutics. The similarities we observe between tumor cells and GN development suggest that the current understanding of human cerebellar development can be leveraged for target identification for differentiation therapies. Extracellular factors like CNTN1, glutamate, and taurine promote differentiation and migration during canonical GN development. Unfortunately, there are no established models of MBEN tumors for us to use in this study, but future experiments can test whether these molecules can also induce differentiation in SHH MBs.

In summary, this work characterizes the differentiated cells in SHH MBs, establishes the key transcriptomic and metabolomic patterns of those cells, and uses these findings to help explain the biological basis of observed molecular subtypes and histological patterns. It is unlikely that this study includes every malignant cell type related to SHH MB biology, but we do show that most tumor cells in pediatric SHH MBs can be associated with some stage of GN development. We hope that these findings promote further research into connections between SHH MB tumorigenesis and cerebellar GN development and that a deeper understanding of this relationship will ultimately enable novel therapeutic approaches.

## Methods
### Material data collection, inclusion, and ethics
All experiments in this study involving human tissue or data were conducted in accordance with the Declaration of Helsinki. All tissues

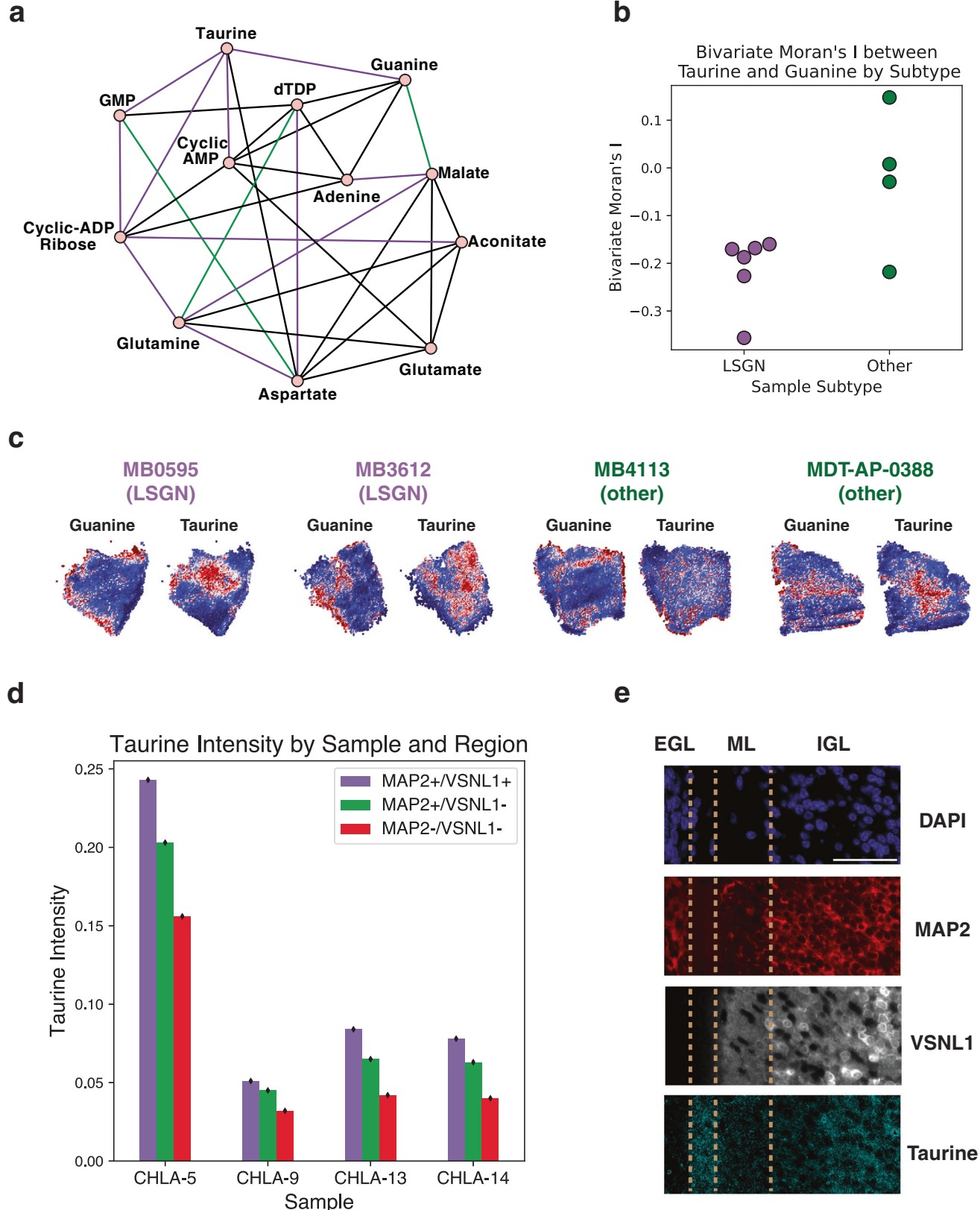

used in this study were obtained with properly informed written consent of patients or their legal representatives.

For MBEN single-cell transcriptomics analysis, all collection and experimental procedures were performed after approval by the institutional review board at The Hospital for Sick Children (Toronto, Canada). For snRNA-seq of samples with known proteomic subtypes, this study was approved by the Institutional Clinical Research Board of Gustave Roussy, and complied with the reference methodology MR-004 (IRB number: 2022-125). For mIHC and imaging analysis of samples from CHLA, all samples were deidentified and obtained with properly informed consent. This study was approved by the Institutional Review Board at

**Fig. 6 | Metabolic features of differentiation in SHH MBs. a** *Select Edges from Joint Graphical Lasso Analysis.* Purple edges appear in at least 50% of the networks from tumors with late-stage GNs, but not for the other samples. Green edges are in at least 50% of other tumor networks, but not samples with late-stage GNs. Black edges appear in networks from 50% of both groups. **b** *Swarmplot of Bivariate Moran's I Between Taurine and Guanine.* Bivariate Moran's I statistic was calculated between the guanine and the spatial lag of taurine for each section. Tumors with late-stage GNs (purple) have a strong negative relationship not consistently observed in the other samples (green). **c** *Taurine and Guanine Anticorrelate in Tumors with Late-Stage GNs.* For four sections, the relative expression values are plotted for guanine and taurine. Each plot shows the metabolite values, clipped at the 3rd and 97th percentiles. The tumors with late-stage GNs show clear spatial anticorrelation between guanine and taurine, which is not consistently observed in the other tumors. **d** *Taurine Intensity by Region Type.* For each of the four samples

(CHLA-5, CHLA-9, CHLA-13, and CHLA-14), the imaged area was divided into three region types: MAP2 + /VSNL1 + , MAP2 + /VSNL1-, and MAP2-/VSNL1-. The height of each bar indicates the mean taurine fluorescence intensity for pixels in each region of that sample. In all four samples, the mean taurine intensity is highest in the MAP2 + /VSNL1+ regions, followed by MAP2 + /VSNL1- and then MAP2-/VSNL1-. Error bars indicate standard error. Data distributions can be found in Supplementary Fig. 24. **e** *Vertical Layers of Pseudo-Cerebellar Structure from CHLA-10.* mIHC from one tumor region from sample CHLA-10. Zoomed in region from orange box in Supplementary Fig. 25. Furthest left region resembles EGL with MAP2 + /VSNL1- cells. On the far right is a region similar the IGL that contains MAP2, VSNL1, and taurine. In between is a region divided in two, where the right section contains VSNL1+ axons and the left one stains for taurine. There appears to be a layer with high taurine levels between the pseudo-ML and pseudo-EGL. Scale bars represent 50 μm.

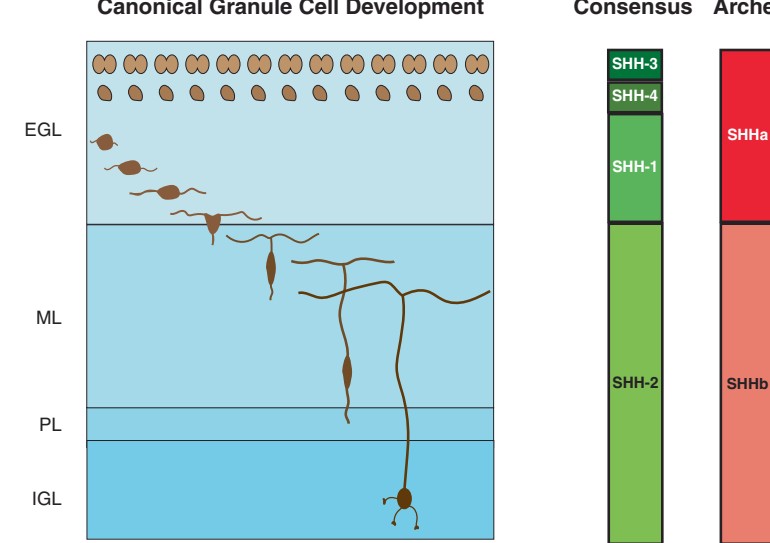

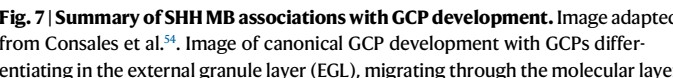

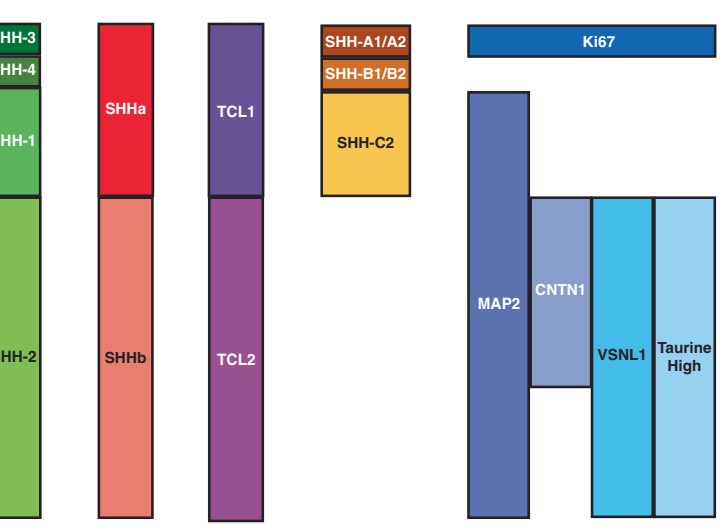

**Fig. 7 | Summary of SHH MB associations with GCP development.** Image adapted from Consales et al.[54]. Image of canonical GCP development with GCPs differentiating in the external granule layer (EGL), migrating through the molecular layer (ML) until they reach their final location in the internal granule layer (IGL). The right side shows medulloblastoma subtypes or IHC staining patterns and their associations with specific regions of the developing cerebellum.

Children's Hospital Los Angeles (CHLA-20-00588). FFPE material for the other samples used for mIHC analysis was obtained with informed consent based on the International Cancer Genome Consortium (ICGC) guidelines. This was approved by the Ethics Committee of the Medical Faculty at Heidelberg University and by the Institutional Review Board of Contributing Center Nikolay Nilovich Burdenko Neurosurgical Institute in Moscow.

### Preparation of single-cell suspensions
Fresh patient tumors were collected at the time of surgical resection. Tumor tissue was mechanically dissociated followed by collagenase-based enzymatic dissociation as previously described in ref. 6.

### Preparation of single-nuclei suspensions
Nuclei were isolated from fresh, snap frozen tumor tissues as previously described in ref. 72. Frozen tissues were dounced in 1 ml of chilled lysis buffer (lysis buffer; 10 mM Tris-HCl (pH 7.4), 10 mM NaCl, 3 mM MgCl$_2$, 0.05% NP-40 detergent) 5 times with a loose pestle, 10 times with a tight pestle and lysed for 10 min on ice. 5 ml of chilled wash buffer (wash buffer; 5% BSA, 0.04 U/μL RNase inhibitor, 0.25% glycerol) was added to the sample, passed through a 40 μm cell strainer and centrifuged at 500 x g for 5 min at 4 °C. After pelleting, the nuclei were resuspended in 5–10 ml of wash buffer. After two washes, single-nuclei suspensions were passed through a 20 μm cell strainer, pelleted, and resuspended in PBS with 0.05% BSA.

### Single-cell and single-nuclei RNA library preparation and sequencing
Single-cell and single-nuclei suspensions were assessed with a trypan blue count. For each sample, 10,000–15,000 cells or nuclei were loaded using the Chromium Controller in combination with the Chromium Single Cell 3' V3 and V3.1 Gel Bead and Chip kits (10X Genomics). Individual cells or nuclei were partitioned into gel beads-in-emulsion (GEMS), followed by reverse transcription of barcoded RNA and cDNA amplification. Individual single-cell libraries with indices and Illumina P5/P7 adapters were generated with the Chromium Single Cell 3' Library kit and Chromium Multiplex kit. The libraries were sequenced on an Illumina Novaseq6000 sequencer. The six additional snRNA-seq samples from the validation cohort were processed using the same protocol and were also run on an Illumina Novaseq6000 sequencer.

### Human medulloblastoma tissue collection (CHLA)
De-identified medulloblastoma samples were retrieved from the Children's Hospital Los Angeles Pediatric Research Biorepository with adherence to the institutional ethical regulations of Children's Hospital Los Angeles Institutional Review Board. Samples retrieved were originally obtained from surgical resection specimens and either frozen or fixed in formalin within 30 minutes of procurement. Frozen samples were embedded in OCT (Tissue Tek) and maintained in monitored freezers at −80 C. All specimens were evaluated by a neuropathologist (J.A.C.) prior to preparation for experimental use.

## VSNL1 immunohistochemistry

VSNL1 staining was performed on a Ventana BenchMark Ultra (Ventana Medical Systems, Tuscon, AZ) on 4-micron sections of paraffin-embedded medulloblastoma tissue. Briefly, slides were deparaffinized and underwent antigen retrieval protocol with cell conditioning 1 (CC1), followed by application of primary antibody (VSNL1, mouse monoclonal, OTI4A6, #MA5-26516, Thermo Fisher Scientific Inc.), at a dilution of 1:1600. 3,3'-diaminobenzidine (DAB) chromogen (ultraView Universal DAB detection kit, Ventana Medical Systems, Tucson, AZ) was used for visualization and counterstaining with hematoxylin was performed.

## Multiplexed immunohistochemistry

FFPE slides of medulloblastoma samples were processed before antibody staining by performing baking, deparaffinization, antigen retrieval, tissue permeabilization, autofluorescence photobleaching, and background imaging with DAPI staining. Slides were baked at 60 °C for one hour. Deparaffinization was completed with the following washes, each at three minutes: xylene (2x), 1:1 ratio of xylene to ethanol, ethanol (2x), 95% ethanol, 70% ethanol, 50% ethanol, RODI water (2x). Slides were submerged in 1x Tris-EDTA and heated in boiling water under pressure of a pressure cooker, then washed in 1x PBS for 10 min. The tissue was permeabilized with 1x PBS + 0.1% TritonX-100 for 10 min. Slides were washed in three changes of 1x PBS + 0.1% Tween20 for 10 minutes/wash.

Autofluorescence photobleaching was carried out by submerging slides in a solution of 4.5% hydrogen peroxide, 25 mM sodium hydroxide, and 1x PBS, while heating the samples and solution to 37 °C and exposing the tissue to direct full visible spectrum LED light. Slides were then washed three times in 1x PBS + 0.1% Tween20 for 10 minutes/wash.

Endogenous non-target proteins were blocked with a one-hour wash of Odyssey Blocking Buffer (PBS). Slides were then washed (1x PBS + 0.1% Tween20, 10 min), incubated in DAPI for 10 min, and then washed again (1x PBS + 0.1% Tween20, 10 min).

Glycerol (10% in 1x PBS) was used to mount coverslips onto the slides for imaging. Background imaging was captured with the same excitation and emission settings as was later used for fluorescent antibody imaging: (ex,em); D360/40x, ET460/50 M; HQ480/40X, 535/50 M; 560/40X, D630/60 M; 628/40X, 692/40 M.

Following background autofluorescence imaging with DAPI staining for registration, samples were incubated in the unconjugated primary antibodies AB1, AB2, AB3 overnight at 4 °C. Samples were then washed three times in 1x PBS + 0.1% Tween20 for 10 minutes/wash. Secondary antibodies with their respective fluorophores were added for one hour at room temperature in a dark humidity chamber.

Fluorescently-labeled tissue slides were imaged on a TE2000 inverted microscope using the excitation emission filters described above with 10x magnification and 0.30 NA lens with a resolution of 1.546 pixels/μm.

After the final round of IHC imaging, slides were photobleached and then stained with hematoxylin and eosin. The slides stained with H&E were imaged on an Aperio AT2 slide scanner at 40x magnification.

## Antibody validation and panels

Antibodies were validated using reference tissues (Supplementary Fig. 26). Two mIHC panels were used for this study and are described below. Panel 1 includes taurine and was run on FFPE tissue from CHLA-5 and CHLA-10. The other mIHC images were generated using Panel 2. MAP2, VSNL1, and Taurine were detected using secondary antibodies. Ki67 was directly conjugated to AF647 before purchasing, while CNTN1 was directly conjugated after purchase using the AF555 kit from Abcam (ab269820).

These antibodies allow for detecting stages of GN development. Ki67 is marker of the cycling progenitor cells[73,74] and MAP2 is expressed by postmitotic granule neurons[75]. CNTN1 is a cell surface marker localized the GN dendrites and axons during development, but the dendritic expression is lost as the GNs mature[76]. VSNL1 is a calcium-sensor expressed in GNs[77] and the snRNA-seq data indicates it expressed exclusively in late-stage GNs.

The antibodies and their dilutions are as follows:

CNTN1: Novus-AF904 (15 μg/ml)

Ki67: Cell Signaling 12075, directly conjugated AF-647 by manufacturer (1:50)

MAP2: Abcam ab92434 (1:100)

Taurine: Sigma-Aldrich AB5022 (1:100)

VSNL1: Invitrogen MA5-26516 (1:100)

The two panels used are as follows:

Panel 1:

Round 1: DAPI, MAP2 (488), VSNL1 (555), Taurine (647)

Round 2: DAPI, CNTN1 (555), Ki67 (647)

Panel 2:

Round 1: DAPI, MAP2 (488), VSNL1 (647)

Round 2: DAPI, CNTN1 (555), Ki67 (647)

## MALDI slide preparation and matrix coating

Fresh tissues were harvested and flash-frozen on dry ice, then stored at 80 °C. 10 μm-thin tissue sections were cut on a cryostat (Leica CM3050S, Wetzlar, Germany) in serial sections for MALDI and IF. Tissue sections were thaw-mounted on indium tin oxide (ITO)-coated glass slides (Bruker Daltonics, Bremen, Germany) and desiccated under vacuum for 10 min before matrix coating.

Slides were subsequently sprayed with negative ionization matrix N-(1-Naphthyl) ethylenediamine dihydrochloride (NEDC, Sigma #222488) using an HTX TM-Sprayer (HTX Technologies, LLC). Concentration of 10 mg/ml was used in 70% MeOH. The sprayer parameter used was 80 °C temperature, 0.1 ml/min flow rate, 1000 mm/min velocity, 2 mm track spacing, 10 psi pressure, and 3 liters/min gas flow rate. Slides were coated on the same day as MALDI imaging.

## MALDI imaging

For MALDI-FTICR scanning, the matrix-coated slides were immediately loaded into a slide adapter (Bruker Daltonics, Bremen, Germany) and then into a solariX XR FTICR mass spectrometer with a 9.4 T magnet (Bruker Daltonics, Bremen, Germany) with resolving power of 120,000 at $m/z$ 500. The laser focus was set to 'small,' and the $x$-$y$ raster stepsize of 50 μm was used using Smartbeam-II laser optics. A spectrum was accumulated from 200 laser shots at 1000 Hz. The ions were accumulated using the cumulative accumulation of selected ions mode (CASI) within an $m/z$ range of 70–300 Daltons before being transferred to the ICR cell for a single scan.

## Hematoxylin and Eosin (H&E) staining for MALDI slides

Histological staining was performed on the same slides after MALDI using Meyer and Briggs' Hematoxylin (Sigma #MHS32) and Eosin (Sigma #HT110332). Matrix was washed off and slide was fixed with cold MeOH for 5 min, washed with PBS 3 times and water, then submerged in hematoxylin for 15 min. Slides were transferred into warm water for 15 min, then dehydrated in 95% EtOH for 30 s, incubated with eosin for 1 min, then dehydrated again stepwise with 95% for 1 min and 100% EtOH for 1 min, and cleared using Xylene for 2 min. Slides were mounted using Cytoseal 60 mounting media (Thermo Scientific #8310-4) and imaged and visualized using a Hamamatsu Nanozoomer with NDP.view2 software (Hamamatsu).

## Statistics and reproducibility

Choosing samples for this study was limited by the fact that medulloblastoma tumors are exceedingly rare. Therefore, no statistical tests were used to determine the sample sizes for this study. The samples used for snRNA-seq, MALDI, and mIHC were chosen purposefully

based on their known histological subtypes and tumor availability. Due to the scarcity of tumor tissue and our focus on histological variability, sex/gender was not considered in sample selection.

Statistical tests performed on the generated omics data are described in the sections below. All code and data used to perform these analyses can be found at https://github.com/fraenkel-lab/shh-mben.

mIHC was run on 8 samples with known proteomic subtypes (4 SHHa and 4 SHHb) and Fig. 5a shows one featured region from 6 tumors with clear nodular structures highlighting the VSNL1 +/− trend. The raw images contain many additional examples for each tumor. Figures 5c, d, & 6d highlight a single region in one sample (CHLA-10) that strongly resembles the developing cerebellum. There are multiple other instances of similar structures within this sample.

### Sample summary
Tumors analyzed for single-cell or imaging analysis are summarized in Supplementary Data 1. Histology information from tumors from Archer et al.[23] was updated based on personal communication with Andrey Korshunov.

### snRNA-seq and scRNA-seq data processing
Quality control was performed for each tumor individually. The filtered counts matrix was used to create a *Seurat* object (version 4.1.0)[78], removing any genes present in less than five nuclei and any nuclei that contain less than 300 features. Nuclei were removed from the dataset if they met any of the following criteria: below 5th percentile of UMI or features detected, above 95th percentile of UMI or features detected, mitochondrial genes represent more than 5% of the counts, or in the top 10% of DoubletScore determined by *Scrublet*[79]. These strong filters left 71,008 high-quality nuclei for analysis.

For the seven snRNA-seq samples, the filtered objects were merged and log-normalized using a scale factor of 10,000. The top 2500 variable genes were identified and the number of UMI counts were regressed out during scaling. Integration was performed using *harmony*[80] and the top 50 dimensions were used for UMAP plotting and Louvain clustering (resolution = 0.25). Non-malignant cells were annotated using marker genes (Supplementary Fig. 1A). Tumor cells were identified by their clustering pattern and expression of known SHH MB genes[8]. Additionally, Gashemi et al. observed similar cell types in their cohort and detected copy number variations in these cells providing further evidence of their malignant status (personal communication). Clusters 6 and 12 were not included in re-clustering of malignant cells because each cluster was primarily associated with a single sample (MB4113 for 6 and MB2112 for 12) (Supplementary Fig. 1B).

Supplementary Fig. 2 details the re-clustering of the snRNA-seq MBEN samples. To generate this plot, nuclei in clusters 0,1,2,3,4 or 5 were selected and the first 50 harmony dimensions were used to calculate nearest neighbors and a UMAP plot. Louvain clustering was then performed on these cells with a resolution of 0.25. Almost every new cluster could be associated with a developmental stage using known marker genes (Fig. 1d). Cluster 6 was merged with cluster 0 to represent the postmigratory GN. For cluster 4, the top marker genes are related to ribosomes. It is unknown whether this cluster corresponds to stage of GN development, so it was excluded from development-related figures. The high ribosomal content indicates these cells could be a sequencing artifact, but similar ribosomal cells are found in our scRNA-seq MBEN sample, the SHH-B2 cells from the Riemondy cohort[8], and the P14 mouse cerebella[6] (Supplementary Fig. 5) suggesting they may be biologically relevant.

The scRNA-seq datasets were processed using similar filters and parameters. The same quality control metrics were used, except the mitochondrial percentage filter was set to 25%. The two scRNA-seq datasets (one MBEN tumor and the P14 mouse dataset from GSE118068) were each analyzed by themselves, so no integration methods were used. Instead, these samples were log-normalized using a factor 10,000 and scaled, and then dimensionality reduction was performed using Principal Component Analysis on the 2500 most variable genes. The top 40 PCs were used for clustering and UMAP plots. For the SHH MBEN tumor, malignant cells were identified by their clustering pattern and expression of SHH MB markers, and these cells were re-clustered using a resolution of 0.2 (Supplementary Fig. 3B). For the P14 mouse, the granule neuron lineage was identified using marker genes and these cells were re-clustered with a resolution of 0.3 and annotated using known markers of GN development (Supplementary Fig. 5).

### Pseudotime analysis
Pseudotime analysis was performed using *monocle3*[81] with a minimal branch length of 15 cells. The root node was set by selecting the nuclei from the cycling GCP cluster (cluster 5) with the highest value for UMAP component 2.

### Plotting with Seurat
Plots from *Seurat* were made using the DimPlot and FeaturePlot functions, adding on specific parameters from *ggplot2*[82] that are described in the code. For many feature plots, the baseline plot was re-made with a custom function that orders the cells by their expression of the relevant gene to help ensure that cells expressing a given marker are visible in the plot. Some feature plots use a minimum cutoff to highlight cells with high expression and these instances are described in the figure legends.

### Clustering of gene set signature scores
Gene set signatures were created for 8 datasets (Supplementary Fig. 4): Archer proteomic subtypes[23], Consensus SHH MB subtypes[5], Korshunov MBEN transcriptional subtypes[24], human GN development[83], MBEN snRNA-seq, MBEN scRNA-seq, P14 Mouse scRNA-seq[6] and Riemondy SHH MB scRNA-seq[8]. For the snRNA-seq and scRNA-seq MBEN datasets, marker genes were identified using the *Seurat* FindMarkers function with default parameters. For these cases, down-sampling was performed before differential analysis so that each cluster was represented by an equal number of cells. The same procedure was used for the P14 mouse to identify the top differential genes for each stage of granule neuron development. Mouse genes were mapped to human orthologs using the *biomartR* package[84]. In a small number of instances, genes were not in the BioMart database. In those cases, if the capitalized version of the mouse gene was present in the MAGIC cohort transcriptomic data, that gene was also included. Otherwise, the gene was not included and the next highest-ranking gene took its place in the signature.

For the consensus subtypes, the transcriptomic microarray data from Cavalli et al.[5] was rank normalized and marker genes were identified by performing t-tests between the subtype of interest and the other SHH tumors[5]. For the Archer et al. subtypes, t-tests were used to identify the differential proteins between SHHa and SHHb[23]. The markers for the Riemondy single-cell clusters and Korshunov et al. MBEN subtypes were taken from their respective supplemental materials (Supplementary Table 3 for Riemondy et al.[8] and Supplementary Tables 1 & 2 for Korshunov et al.[24]).

From these marker gene lists, a gene signature was created by taking the top n genes (50, 100, 150, & 200). In some cases, n was larger than the number of marker genes for published datasets. When this occurred, all marker genes were included for the given signature. These signatures are summarized in Supplementary Data 2.

All genes sets of size n were used together as inputs for Gene Set Variation Analysis (GSVA), which takes in gene-level data and calculates enrichment scores for each gene set[45]. GSVA was run on transcriptomic data from the 223 SHH tumors of the MAGIC cohort[5]. This approach

allows for investigating the relationships between many gene sets because activation scores are generated on a single cohort of representative SHH tumors.

*ConsensusClusterPlus* was used to run consensus clustering on the activation scores to better understand what signatures are activated in the same samples[85]. This method repeatedly subsamples items (i.e. gene signatures) and features (i.e. SHH tumors) and clusters the data to determine which signatures cluster together. 1000 sub-samplings were run using the k-means algorithm and Euclidean distance metric. For each run, all of the gene signatures were used, but the SHH samples (i.e. features) were subsampled for the following percentages: 30%, 50%, 70%, and 90%. For each parameter set, the method was run from $k = 2$ to $k = 10$ and the optimal k was found to be 5 clusters for all parameter sets using an elbow plot.

The consensus clustering plots were made using the clustermap function in *seaborn*[86] using correlation as the distance metric. All gene set sizes and subsampling percentages revealed similar high-level trends (Supplementary Fig. 6) with four primary groups representing developmental stages: cycling progenitors, non-cycling progenitors, premigratory GN, and migrating/postmigratory GN (late-stage GNs).

### Copy number variation (CNV) analysis

Copy number variation data for chromosomal arms was included in the MAGIC cohort data from Cavalli et al.[5]. Two-sided Mann-Whitney U tests were performed to determine associations between CNVs and GSVA gene set signatures from MBEN snRNA-seq cohort. The following gene sets were considered: Cycling GCP, GCP, Ribosomal, Premigratory GN, Migrating GN, Postmigratory GN, Proliferation (Cycling GCP score – GCP score), Progenitor (Cycling GCP score + GCP score), and Late-Stage GN (Migrating GN score + Postmigratory GN score). For each CNV/signature pair, two Mann-Whitney U tests were performed. First, all samples with a loss of the chromosome arm were compared with tumors having a wildtype or gain status, and then a second test was performed to compare the samples with a gain to the other tumors. If no tumors had a loss or gain, that test was not performed, leaving 657 comparisons. Multiple associations were significant using the Bonferroni-corrected alpha of 1.52 E −5. These analyses were performed using gene set signature scores based on 100 genes, but the relationships shown in Fig. 3c are also significant for signatures of size 50, 150, and 200. All comparisons are detailed in Supplementary Data 4.

### Post-transcriptional regulation analysis

Proteomic and transcriptomic data from Archer et al.[23] were re-analyzed, considering the 8674 genes identified in all tumors for both assays. For each sample, the protein and RNA values were rank-normalized from −0.5 to 0.5 and a rank difference (protein rank – RNA rank) was calculated for each gene as a proxy for post-transcriptional regulation. Thus, a gene with a high protein rank in a given sample and a low RNA rank would have a strong positive rank difference, suggesting possible post-transcriptional upregulation. Genes were categorized as synaptic if they appear in the GO_SYNAPSE gene set from MSigDB[87] and as FMRP targets if they appear in the stringent list in Supplementary Fig. 2A from Darnell et al.[48].

For SHHb analysis, the mean protein rank, RNA rank, and rank difference were calculated for every gene by taking the average value for the five SHHb tumors. The same procedure was applied to the nine SHHa tumors for SHHa analysis. Synaptic and non-synaptic gene ranks were compared using a two sample t-test in *scipy*[88]. To determine potential post-transcriptional regulation, rank differences were compared to mean of 0 using a one-sample t-test.

### snRNA-seq analysis for validation cohort

snRNA-seq was performed on six additional tumors with known proteomic subtypes (personal communication with Olivier Ayrault).

MB002, MB009, and MB019 are classified as SHHa proteomic subtype, while MB005, MB015, and MB084 are called as SHHb. The output data was processed using the same parameters and quality control as the original MBEN cohort and then combined with the original MBEN cohort to perform clustering and feature analysis on one large cohort. *Harmony*[80] was used for integration and Supplementary Fig. 8 shows the resulting UMAP plots for the new samples.

### Processing of published scRNA-seq and snRNA-seq

Data for figure for Supplementary Figs. 9, 10, 11 and 14 were generated by reprocessing data from published scRNA-seq studies[7,8]. In each case, UMAP plots were generated by reprocessing the counts data from each study and displayed using the FeaturePlot function from *Seurat*. Supplementary Figs. 12 and 13 were generated using R data objects provided by the authors from Vladoiu et al. (2019)[6].

### Image registration and processing

Individual images were taken with 10% horizontal and vertical overlap and then stitched together with the Microscopy Image Stitching Tool (MIST)[89] through *FIJI*[90]. Image registration between mIHC rounds was performed using the *MultiStackReg* plugin in *FIJI*[90] using the DAPI channels and the rigid body transformation. When the two images were not identical sizes, the two images were cropped slightly around the edges to enforce identical sizing. In a small number of select images, there were clear visual artifacts with extremely high fluorescence. In such cases, these regions were not considered for downstream analysis.

### MALDI data processing

MALDI intensity data was analyzed for 4677 consensus $m/z$ peaks identified by the *Isoscope* package[91]. For each $m/z$ value in each MALDI spot, the intensity was assigned to the maximum intensity of peaks within 2 PPM of the $m/z$ value. These data were then loaded into a *scanpy* object[92], and each spot was normalized using the total ion count (TIC) method, whereby every $m/z$ intensity is divided by the sum of the intensities for that spot. The data was then scaled, and dimensionality reduction was performed using PCA. The spots were clustering using Leiden clustering with a resolution of 0.2 and considering the 10 nearest neighbors and first 10 PCs. Some clusters (2,4,9,10,11,12,13, &14) were primarily around the edges and thus removed as they were likely artifacts. Additionally, there were clusters of spots in each sample that were visible artifacts compared to H&E staining and these clusters were removed as well. This resulted in 52,393 high-quality spots that strongly resemble the tissue structure from H&E stains (Supplementary Fig. 22).

Our analysis was focused on high-quality annotated metabolites. First, $m/z$ peaks were associated with known metabolites using the database from Supplementary Data 5 and a window of 2 PPM. These features were then further filtered to only include metabolites that have an intensity greater than 100,000 in more than 50% of the MALDI spots. These cutoffs produced 56 high-quality metabolites for network and correlation analysis. To assess the robustness of these results, this analysis was repeated using RMS normalization in the *Cardinal* package[93]. This investigation showed the same trend between taurine and guanine (Supplementary Fig. 23) and that taurine had a stronger centrality in tumors with LSGNs compared to others.

### Joint graphical lasso

Joint graphical lasso[61] analysis was performed using the *gglasso*[94] python package (version 0.1.9). Graphical lasso[95] uses observational data to learn a sparse approximation of the precision matrix, where each 0 represents two metabolites whose expression levels are independent of each other when considering the intensities of the other metabolites. This leaves of matrix of non-zero values, which can be represented as a network with edges between conditionally dependent

metabolites. Joint graphical lasso allows for approximating multiple precision matrices at one time, while sharing information between the related datasets.

The group graphical lasso algorithm was used for the TIC-normalized MALDI for the 56 annotated metabolites. A parameter sweep was performed for lambda1, which controls sparsity, and lambda2, which promotes similarity across networks, using the following parameters: lambda1 (0.05, 0.1, 0.15, 0.2, 0.25) and lambda2 (0.01, 0,02, 0,03, 0,04 and 0.05). The optimal parameters (lambda1 = 0.1, and lambda2 = 0.01) were the minimal empirical bayes score, which was calculated using the ebic function from the *gglasso* package with a gamma parameter of 20 to promote sparser networks. Betweenness centrality was calculated for the output network for each sample using the *networkx* python package (version 2.6.3). Cytoscape (version 3.7.2)[96] was used to create Fig. 6a, showing edges present in particular samples.

### Bivariate Moran's I analysis
Bivariate Moran's I analysis was performed using the *pysal* (version 2.4.0) package to analyze what metabolites correlate with the spatial lag of taurine. Weights were determined using the 24 nearest spots (i.e. two levels out on the MALDI spot grid) and then row normalized. The bivariate Moran's I statistic was calculated between guanine and taurine for each spot using the Moran_BV function.

### Taurine focused immunohistochemistry experiment
To investigate how taurine staining relates to MAP2 and VSNL1, immunohistochemistry was performed on 11 SHH MBs from CHLA (Supplementary Data 1). Visual examination revealed four samples with significant variability in VSNL1 and MAP2 staining: CHLA-5, CHLA-9, CHLA-13, and CHLA-14. For these cases, a small 3 × 3 or 4 × 4 region was chosen for imaging because it contained MAP2 + /VSNL1 + , MAP2 + /VSNL1-, and MAP2-/VSNL1- areas.

Imaging was performed on a Zeiss LSM980 microscope using the Zen 3.6 software package. For each sample, a 10548 × 10548-pixel tiled image representing 1.57 mm$^2$ was captured (4 × 4 tiles, 10% overlap, 1 px = 0.149 μm$^2$). A Plan-Apochromat 20x/0.8 M27 air objective was used for all images with a GaAsP-PMT detector. Stitching of tiles was performing using the default settings in the Zen 3.6 software (10% tile overlap, 488 channel used as reference).

### Taurine focused image processing
Nuclei were identified in the DAPI channel for each image by using *CellProfiler* (version 4.2.4)[97]. *CellProfiler* was also used to quantify the intensity of nuclear markers, like Ki67, and to generate coordinates for each cell. To identify regions that have positive signal for the non-nuclear markers (i.e. VSNL1 and MAP2), each channel was first gated using Gaussian mixture models (GMMs) from scikit-learn (version 1.2.1)[98]. For each channel, the GMM was applied to the log-transformed images to identify three components: background, negative signal, and positive signal. The initial gating threshold was calculated as the average of the means of the second and third components. Based on visual inspection, the gates were manually adjusted to reflect the positive signal. To capture continuous regions of positive signal, three scikit-image[99] (version 0.21.0) functions were implemented to generate masks: isotropic_closing, remove_small_holes, and remove_small_objects.

### Ki67 quantitative analysis
Cells were labeled as positive or negative for Ki67 expression using the quantification values from *CellProfiler* (version 4.2.4)[97]. The values were gated using thresholds calculated using Gaussian mixture models (GMMs). The GMMs were applied to log transformed Ki67 values to identify positive and negative signal and this gate was manually adjusted to match visual inspection of the images. To compare the Ki67 positivity rate in VSNL1+ and VSNL1- areas, masks were generated for VSNL1 staining in MB287 and CHLA-5 using the same procedure described for the taurine-focused analysis (Supplementary Fig. 27). Any cell located within the boundaries of a given mask was considered positive for that marker. A Fisher's exact test was applied to determine whether Ki67 status was independent of VSNL1 status.

### Taurine quantitative analysis
Region masks were created for each image so that nodules could be classified as VSNL1 + /MAP2 + , MAP2 + /VSNL1-, or negative for both markers. Additionally, each mask was intersected with a tissue mask, generated from DAPI positive regions. In each image, the taurine intensity was calculated for every pixel in each region type. Differential analysis was performed by comparing the intensity of taurine in the MAP2 + /VSNL1+ regions with the MAP2 + /VSNL1- nodules.

### Reporting summary
Further information on research design is available in the Nature Portfolio Reporting Summary linked to this article.

## Data availability
The raw scRNA-seq and snRNA-seq data generated for this manuscript have been deposited in GEO under the accession number GSE214469. The previously published data used for this work can be accessed through the following GEO accession numbers: scRNA-seq data from Riemondy et al.[8] GSE156053, scRNA-seq data from Hovestadt et al.[7] GSE119926, scRNA-seq data from Vladoiu et al.[6] GSE118068, and bulk RNA transcriptomics from Cavalli et al.[5] GSE85218.

mIHC imaging data can be found at https://zenodo.org/records/10257144 and taurine-focused IHC imaging can be found at https://zenodo.org/records/10256482. Processed omics data can be found with the corresponding code at https://github.com/fraenkel-lab/shh-mben. The remaining data are available within the Article, Supplementary Information, or the Source Data file. Source data are provided with this paper.

## Code availability
Custom R and python scripts for data processing, analysis, and presentation can be found on github at https://github.com/fraenkel-lab/shh-mben.

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

## Acknowledgements

This study was funded by U01-CA184898 (E.F, J.P.M., S.L.P.), U01-CA253547 (J.P.M., E.F., R.W.R.), R01-NS089076 (E.F.), P50-HD105351 (S.L.P.), R35-NS122339 (R.W.R.), U24-CA220341 (J.P.M.), R01-NS106155 (M.D.T.), R01-CA159859 (M.D.T.), R01-CA255369 (M.D.T.), SU2C-AACR-DT1113 (M.D.T.), SU2C-AACR-DT-19-15 (M.D.T.), SU2C - Convergence 3.14 (S.M.D.), M.I.T. Takeda Fellowship (M.P.G.), Robert J. Arceci Innovation Award from St. Baldrick's Foundation (OA), PEDIAC consortium, INCA_15670 (OA), and INCa PRTK-19-027 (CD/OA). Additionally, M.D.T. is supported by The Pediatric Brain Tumour Foundation, The Terry Fox Research Institute, The Canadian Institutes of Health Research, The Cure Search Foundation, Matthew Larson Foundation (IronMatt), b.r.a.i.n.-child, Meagan's Walk, SWIFTY Foundation, The Brain Tumour Charity, Genome Canada, Genome BC, Genome Quebec, the Ontario Research Fund, Worldwide Cancer Research, V-Foundation for Cancer Research, and the Ontario Institute for Cancer Research through funding provided by the Government of Ontario, Canadian Cancer Society Research Institute Impact grant, a Cancer Research UK Brain Tumour Award, and the Garron Family Chair in Childhood Cancer Research at the Hospital for Sick Children and the University of Toronto. D.R.G. was supported with personal grants by the German Academic Scholarship Foundation (Studienstiftung des Deutschen Volkes) and the Mildred Scheel Doctoral Fellowship program of the German Cancer Aid (Deutsche Krebshilfe). DRG and KWP are thankful to the non-profit foundation Ein Kiwi gegen Krebs for their support. We also thank Yoon-Jae Cho, John Michaels, Koei Chin, Joe Gray, Connie New, and Ali Abdullatif for their help with the manuscript. Additionally, we appreciate support from the USC Norris Comprehensive Cancer Center Translational Pathology Core (P30CA014089), the Pediatric Research Biorepository at CHLA, and the Histology Core at the Koch Institute at MIT.

## Author contributions

M.P.G. and E.F. led the study. W.O., M.C.V., and N.C.H. generated snRNA-seq data. M.C.V. and L.K.D. generated scRNA-seq data. A.M.M., J.B., & J.A.G. performed antibody validation and A.M.M. and A.D. ran mIHC experiments. J.A.G. ran the VSNL1 IHC and A.D.W. anonymized CHLA data. N.C.H., V.P., and M.P.G. performed mIHC image analysis. S.M.D. and N.R.P. generated MALDI-IMS data. M.P.G. performed data analysis and developed the gene signatures method. R.A.Sa. assisted with data and sample organization for snRNA-seq and MALDI. M.P.G. wrote the initial manuscript draft and M.P.G., E.F., J.A.C., A.M.M., R.W.R., J.P.M., M.D.T., and L.G. helped with editing. M.P.G., J.A.C., V.P., and A.M.M. generated figures. E.F., M.D.T., L.G., R.W.R., S.L.P., J.P.M., O.A., J.A.C., R.A.Se., G.H.H., T.E., D.R.G., and K.R.P. provided medullo-blastoma expertise for data interpretation and experimental design. M.D.T., J.A.C., S.L.P., O.A., C.D., and A.K. provided material for the study. E.F. supervised the study.

## Competing interests

The authors declare no competing interests.

## Additional information

[1]Department of Biological Engineering, Massachusetts Institute of Technology (MIT), Cambridge, MA, USA. [2]Developmental & Stem Cell Biology Program, The Hospital for Sick Children, Toronto, ON, Canada. [3]Department of Laboratory Medicine and Pathobiology, University of Toronto, Toronto, ON, Canada. [4]Hopp-Children's Cancer Center Heidelberg (KiTZ), Heidelberg, Germany. [5]Division of Pediatric Neuro-oncology, German Cancer Consortium (DKTK), German Cancer Research Center (DKFZ), Heidelberg, Germany. [6]Department of Pediatric Oncology, Hematology, and Immunology, Heidelberg University Hospital, Heidelberg, Germany. [7]Department of Pathology and Laboratory Medicine, Children's Hospital Los Angeles (CHLA), Los Angeles, CA, USA. [8]Lewis-Sigler Institute for Integrative Genomics, Princeton University, Princeton, NJ, USA. [9]Department of Molecular Biology, Princeton University, Princeton, NJ, USA. [10]Ludwig Institute for Cancer Research, Princeton University, Princeton, NJ, USA. [11]Department of Cancer Biology, Dana-Farber Cancer Institute, Boston, MA, USA. [12]Department of Neurobiology, Harvard Medical School, Boston, MA, USA. [13]Cancer Genome and Epigenetics Program, NCI-Designated Cancer Center, Sanford Burnham Prebys Medical Discovery Institute, La Jolla, CA, USA. [14]The Arthur and Sonia Labatt Brain Tumour Research Centre, The Hospital for Sick Children, Toronto, ON, Canada. [15]Department of Child and Adolescent Oncology, Gustave Roussy, Villejuif, France. [16]INSERM U981, Molecular Predictors and New Targets in Oncology, University Paris-Saclay, Villejuif, France. [17]Cancer Research Program, McGill University, Montreal, QC, Canada. [18]MUHC Research Institute, McGill University, Montreal, QC, Canada. [19]Herbert Irving Comprehensive Cancer Center, Columbia University Medical Center, New York,

NY, USA. [20]Department of Neurology, Columbia University Medical Center, New York, NY, USA. [21]Department of Medicine, Moores Cancer Center, UC San Diego, La Jolla, CA, USA. [22]Clinical Cooperation Unit Neuropathology (B300), German Cancer Research Center (DKFZ), Heidelberg, Germany. [23]German Cancer Consortium (DKTK), Heidelberg, Germany. [24]Department of Neuropathology, Heidelberg University Hospital, Heidelberg, Germany. [25]Department of Neurology, Boston Children's Hospital, Boston, MA, USA. [26]Harvard Medical School, Boston, MA, USA. [27]Broad Institute of MIT and Harvard, Cambridge, MA, USA. [28]Institut Curie, PSL Research University, CNRS UMR, INSERM Orsay, France. [29]Université Paris-Saclay, CNRS UMR 3347, INSERM U1021 Orsay, France. [30]Rutgers Cancer Institute of New Jersey, New Brunswick, NJ, USA. [31]Division of Pulmonary and Critical Care Medicine, Department of Medicine, The Feinberg School of Medicine, Northwestern University, Chicago, IL 60611, USA. [32]Department of Pathology, Keck School of Medicine, University of Southern California, Los Angeles, CA, USA. [33]Division of Neurosurgery, The Hospital for Sick Children, Toronto, ON, Canada. [34]Department of Surgery, University of Toronto, Toronto, ON, Canada. [35]Department of Medical Biophysics, University of Toronto, Toronto, ON, Canada. [36]Texas Children's Cancer Center, Hematology-Oncology Section, Houston, TX, USA. [37]Department of Pediatrics – Hematology/Oncology and Neurosurgery, Baylor College of Medicine, Houston, TX, USA. ✉e-mail: fraenkel-admin@mit.edu

