## [Peer Review File · Nature Communications]

Reviewers' Comments:

Reviewer #1:

Remarks to the Author:

Gold et al. investigated whether heterogeneity in medulloblastoma tumors was due to differences in developmental pathways. The authors used single-nucleus RNA sequencing on highly differentiated medulloblastoma tumors with Sonic Hedgehog activation. The study shows that malignant cells resembled stages of canonical granule neuron development. The authors further connected results from single-cell profiling to published datasets and found that some established molecular subtypes of SHH MB appeared arrested at different developmental stages. Furthermore, the authors identified distinct metabolic and histological profiles for highly differentiated tumors.

Several comments:

1. The study mentions that differences were observed between snRNA-seq and scRNA-seq results. Could the authors elaborate on this? Could these differences affect the results of the trajectory analysis?
2. The authors identify a novel association between the SHH-C2 signature and premigratory GNs - I think it would be helpful to briefly explain to readers what the SHH-C2 signature is (at least it wasn't clear to this reviewer).
3. p-values are missing in Figures 3 c/d and 4d.
4. Are there biological pathways that are activated in tumor cells that have loss of chr 9q but have high activation scores for late stage GNs? These signatures/pathways may suggest how cells with large CNVs overcome a differentiation block.
5. FMRP promotes nuclear export of its target genes during neural differentiation. Would it be possible to detect this phenomenon in snRNA-seq tumors from SHHb tumors? Perhaps as an FMRP signature?
6. IHCs are missing scale bars.
7. In Fig. 6e, what is the region that shows high taurine staining? Could the authors elaborate?

Reviewer #2:

Remarks to the Author:

Authors analyzed seven sonic hedgehog (SHH) activation Medulloblastoma (MB) tumors to study the heterogeneity of the tumor microenvironment. They combined single nucleus RNA seq, cyclic immunofluorescence, and MALDI metabolic imaging analysis. Overall, MB's molecular heterogeneity is a significant problem, but there are a few concerns:

- 1) Only 7 MBs were analyzed. Can authors expand this to a larger cohort to validate biological findings around developmental trajectories and corresponding metabolic profiles.
- 2) cyclic immunofluorescence datasets are underutilized as the single cell segmented data has not been explored. Authors should incorporate single cell proteomics profiles along with cell type information to benchmark MB tumor differences.
- 3) MALDI data yield a larger number of metabolites. It is unclear how selected metabolites and their networks are identified. Authors should validate, benchmark, and quantify MALDI data and corresponding metabolite annotations carefully.
- 4) Because there are multiple dataset types, how are the data normalizations (z-score and multiomics data integration) incorporated into the signature model shown in Fig 2.
- 5) Figures lack colormaps of trajectory graphs etc. and text labels are too small to read.

While the presented work is promising, the statistical power with n=7 is under-representative of MB

heterogeneity. Data integrations need more rigor. Metabolite annotations need validations. Overall, a major revision will be required to make this paper solid for the journal.

Reviewer #3:

Remarks to the Author:

The manuscript by Gold et al., provides important information on the biological bases of the Sonic Hedgehog medulloblastoma (SHH-MB) heterogeneity. The authors hypothesize that the inter- and intra-tumoral heterogeneity of SHH-MB is dependent from their developmental origin. Through snRNA-seq on seven MBEN tumors and by re-analyzing published data sets, Gold and collaborators find that malignant cells resemble every stage of canonical cerebellum granule neuron development. In particular, they find that each molecular subtype of SHH-MB is characterized by a specific developmental stage and that specific genomic alterations, such as CNVs, are also associated with distinct developmental stages. Finally, the authors identify key metabolic and histological profiles for extremely differentiated tumors.

Although additional validation studies are needed, this work provides novel approaches to investigate the relationship between tumor heterogeneity and developmental programs aimed to better stratify individual patient risk and to identify innovative therapeutic opportunities.

Overall, data from this work will represent a valuable resource for the tumor biology studies.

The manuscript is well written, however, a brief description of the different SHH-MB clusters from Archer et al., Riemondy et al., and Korshunov et al., would make the manuscript easier to read, especially for researchers who are not in the medulloblastoma field.

A figure, similar to figure 1C, reporting SHH-MB subtypes associated with distinct granule development stages, specific biomarkers, CNVs, and metabolites, should be included in the main text.

Given the frequency of CNVs in SHH-MB, has the association with other chromosomal alterations found in childhood SHH-MB (i.e. 17p-, 2+) been investigated?

Line 748, please check the paragraph title.

Point by Point Response

Reviewer 1:

1. The study mentions that differences were observed between snRNA-seq and scRNA-seq results. Could the authors elaborate on this? Could these differences affect the results of the trajectory analysis?

We observed two main differences between snRNA-seq and scRNA-seq. The first was the expression of specific marker genes. For example, SFRP1 was strongly expressed in the GCP-like cells in scRNA-seq but not as strongly in snRNA-seq. Additionally, we saw a difference in the proportion of cells resembling late-stage GNs when they were present. In the snRNA-seq, we found that for the 5 MBEN tumors containing late-stage GNs, the GNs represented 66.3% of the tumor cells, while in the scRNA-seq sample, these cells were only 33.5% of the tumor cells.

We hypothesize that the difference between snRNA-seq and scRNA-seq proportions may be due to neuron-like properties of these tumor cells. Based on expression and imaging data, these differentiated tumor cells express canonical neuronal genes, and neurons are known to be underrepresented in scRNA-seq data (1,2). It is also possible that the differences reflect the one scRNA-seq sample in our cohort. Running additional single-cell RNA-sequencing on MBEN tumors was not possible because these tumors are remarkably rare and fresh cells are needed for scRNA-seq.

We summarized these findings within the manuscript by adding a paragraph to the end of the first results section (page 5).

Since there are known biases in single-cell and single-nucleus sequencing⁴¹, we also performed scRNA-seq on fresh cells from one MBEN tumor (Supplementary Figure 3A). The scRNA-Seq cells do not express the same exact markers as the snRNA-Seq nuclei, but we still observe clusters of malignant cells with markers for each stage of GN development (Supplementary Figure 3B). Additionally, we observed a difference in cell type proportions between scRNA-seq and snRNA-seq. For the five snRNA-seq samples that contain all stages of GN development, 66% of the tumor cells resemble migrating and postmigratory GNs. In the one scRNA-seq sample, these late-stage GNs only represent 33% of the malignant cells. This difference may be due to sample-to-sample variability or technical differences between cells captured by each assay^{42,43}

2. The authors identify a novel association between the SHH-C2 signature and premigratory GNs - I think it would be helpful to briefly explain to readers what the SHH-C2 signature is (at least it wasn't clear to this reviewer).

We appreciate this comment and similar ones made by other reviewers. To address this, we added two paragraphs describing the major subtypes and single-cell clusters identified by the previous studies (page 7).

We considered signatures from six studies related to SHH MB or GN development (Supplementary Figure 4). Cavalli *et al.* ⁵ defined four consensus subtypes from bulk RNA and methylation; SHH-3 (α) and SHH-4 (δ) are associated with older patients, while SHH-1 (β) and SHH-2 (γ) tumors are often more neuronal and come from younger patients. Archer *et al.* ²³ identified proteomic subtypes where SHHa tumors have proliferation and ribosomal markers, while SHHb samples have elevated levels of proteins related to synapses and glutamate signaling. Korshunov *et al.* ²⁴ identified TCL1 and TCL2 transcriptional subtypes of SHH MBs, where TCL2 tumors have high expression of neuronal genes and come from patients with exceptional survival rates.

In addition to these three bulk omics experiments, we also included cell types from three scRNA-seq studies. Riemondy *et al.* (2022) ⁸ analyzed SHH MBs and found six clusters of tumor cells. SHH-A cells are associated with proliferation (SHH-A1 with S phase and SHH-A2 with G2M phase), while SHH-B cells are progenitors, with SHH-B1 resembling GCPs and SHH-B2 having high ribosomal expression. Lastly, SHH-C1 cells are enriched for RNA processing and axo-dendritic transport genes, while SHH-C2 cells have high levels of neural development markers like STMN2. We also generated signatures for cell types found during cerebellar GN development and considered studies from both human ⁴⁴ and mouse ⁶.

3. p-values are missing in Figures 3 c/d and 4d.

Thank you for noticing this. We added the appropriate p-values to the plots (pages 11 & 14) and added further descriptions in the methods.

4. Are there biological pathways that are activated in tumor cells that have loss of chr 9q but have high activation scores for late stage GNs? These signatures/pathways may suggest how cells with large CNVs overcome a differentiation block.

This is an excellent question and something we have been curious about for a while. In the Cavalli *et al.* cohort, there are 7 samples with loss of chromosome 9q and high scores for late-stage granule neurons (all being in the “neuronal” SHH-1 and SHH-2 groups). Every one of these samples have some other CNV, but there are no common CNVs across the tumors.

To investigate this further, we compared the RNA expression profiles between these 7 samples with 9q loss and the expression profiles from the other tumors from SHH-1 and SHH-2 patients. We analyzed the top differential genes using Enrichr, which highlighted a gene set for NGF-stimulated transcription; this is interesting because the NGF receptor (NGFR or p75^{NTR}) is known to also play a role in granule cell precursor exit (3). So, it is plausible that these tumors are exiting through a different mechanism, but we do not believe the hypothesis is strong enough to include in the manuscript.

5. FMRP promotes nuclear export of its target genes during neural differentiation. Would it be possible to detect this phenomenon in snRNA-seq tumors from SHHb tumors? Perhaps as an FMRP signature?

To address this, we considered the FMRP target gene signature used in Figure 4 (4) and scored all of the snRNA-seq cells for these genes. We observed that these RNAs are highest in the late stage GNs (migrating and postmigratory GN) that are found in SHHb. We have now included this result in the text and referenced it in Supplementary Figure 17.

6. IHCs are missing scale bars.

We apologize for not including this in the original version, but this has been fixed. Thank you for pointing this out.

7. In Fig. 6e, what is the region that shows high taurine staining? Could the authors elaborate?

We observe this distinct region of taurine immunoreactivity at the most external part of region resembling the molecular layer in this pseudo-cerebellar structure within an MBEN nodule. We know that taurine is hypothesized to play a role in canonical granule neuron development (5,6) and this regional positivity is present between the pseudo-EGL and the pseudo-molecular layer, where in the developing cerebellum, GNs are actively differentiating. In the tumor context, we do not know what this pattern means, but wanted to describe the observation for other researchers.

In order to further investigate taurine staining patterns, we ran additional immunohistochemistry to assess the taurine levels in three tumor regions: MAP2+/VSNL1+ areas resembling late-stage GNs, MAP2+/VSNL1- areas resembling premigratory GNs, and MAP2-/VSNL1- internodular regions resembling the GCPs. We found higher levels of taurine in the VSNL1+ areas, further suggesting that these extremely differentiated areas have higher levels of taurine (Figure 6d).

Reviewer 2:

1) Only 7 MBs were analyzed. Can authors expand this to a larger cohort to validate biological findings around developmental trajectories and corresponding metabolic profiles.

The rarity of medulloblastoma tumors, and the fact that they are pediatric and located in the brain, means that the worldwide supply of samples is quite limited. To date, scRNA-seq has been published for only 15 SHH medulloblastoma tumors. Nevertheless, we have succeeded in obtaining six additional SHH medulloblastoma tumors and performed snRNA-seq on them (Supplementary Figure 8). Our new cohort covers additional histological subtypes of SHH MB and effectively doubles the number of SHH MB tumors with published single-cell transcriptomics data. In addition, we validated that our observations are supported by the 15 SHH medulloblastoma tumors that have already been published (Supplementary Figures 9-14).

We unfortunately were unable to run additional MALDI, as high-quality tumor slices were not available. To address the key metabolic profile, we performed immunohistochemistry on 11 samples to assess the correlation between taurine and different neuronal markers. We quantitatively analyzed this relationship in four tumors with variable VSNL1 staining and found that taurine intensity is strongest in the VSNL1+ areas (Figure 6d).

2) cyclic immunofluorescence datasets are underutilized as the single cell segmented data has not been explored. Authors should incorporate single cell proteomics profiles along with cell type information to benchmark MB tumor differences.

We thank the reviewer for raising this point. In response, we sought to better utilize the imaging data to quantitatively assess the relationship between Ki67+ and neuronal markers (originally described in Figure 5B). We focused this analysis on two tumors, MB287 and CHLA-5, because they contain significant variability in VSNL1 staining. In these samples, we ran the corresponding cyclF data through cell segmentation and marker quantification pipelines described in the updated methods section on page 33. In both tumors, we observed that the percentage of Ki67+ cells is significantly lower in the VSNL1+ areas compared with the VSNL1- areas. We also intend to make the cyclF data easily accessible so that outside researchers can perform additional analysis if they wish.

3) MALDI data yield a larger number of metabolites. It is unclear how selected metabolites and their networks are identified. Authors should validate, benchmark, and quantify MALDI data and corresponding metabolite annotations carefully.

We apologize if our methods were not clear. The MALDI data processing was performed using the Isoscope package, published by Wang et al. (7) and in collaboration with the Davidson Lab who are experts in MALDI-IMS. The details of the metabolite mapping are in the Methods section titled "MALDI Data Processing". The metabolites were annotated using the Davidson's lab m/s peak database, which is described in Supplementary Table 5. This table indicates which m/z peaks are known to be validated using established wet lab techniques (like isotopes or standards) and which ones are simply based on being within 2 PPM of a known metabolite's m/z range. Taurine was the key metabolite to come out of our MALDI analysis; this metabolite was validated by the Davidson lab through ms/ms and we provided further confirmation by performing immunofluorescence experiments.

To address the reviewer's concerns and ensure the quantitative results are robust, we also performed the entire MALDI data processing again using Cardinal (8), another established computational package. With this pipeline, we still observe strong taurine/guanine anticorrelation in the tumors with late-stage GNs (Supplementary Figure 23) and find that taurine is a hub node in the co-expression network for the LSGN tumors but not the others.

4) Because there are multiple dataset types, how are the data normalizations (z-score and multiomics data integration) incorporated into the signature model shown in Fig 2.

We apologize that this methodology was not clearer in the text and we added sentences to the methods (page 29) to help clarify how all the signatures incorporated in Figure 2 are based on genes from individual experiments with a single type of data. The integration of different data types only takes place after the samples are scored. This approach was specifically designed to integrate many assays at once by only relying on the top differential genes and not using the raw data. Each well-controlled study was analyzed by itself to identify the top markers for each cell type or subtype of interest and then we used GSVA to score each marker gene set on RNA expression data from a large heterogeneous cohort of 223 SHH medulloblastomas. We then analyzed the relationships between the GSVA activation scores to understand what identities are activated in similar samples.

5) Figures lack colormaps of trajectory graphs etc. and text labels are too small to read.

We thank the reviewer for pointing this out. We modified these trajectory graphs to include color maps and increased the font size on many figures to make them more readable.

Reviewer 3:

- 1. The manuscript is well written, however, a brief description of the different SHH-MB clusters from Archer et al., Riemondy et al., and Korshunov et al., would make the manuscript easier to read, especially for researchers who are not in the medulloblastoma field.**

We appreciate this feedback. It was very similar to comments made by other reviewers, so we included an extra paragraph in the second results section of the paper to better inform the readers (page 7).

We considered signatures from six studies related to SHH MB or GN development (Supplementary Figure 4). Cavalli *et al.*⁵ defined four consensus subtypes from bulk RNA and methylation; SHH-3 (α) and SHH-4 (δ) are associated with older patients, while SHH-1 (β) and SHH-2 (γ) tumors are often more neuronal and come from younger patients. Archer *et al.*²³ identified proteomic subtypes where SHHa tumors have proliferation and ribosomal markers, while SHHb samples have elevated levels of proteins related to synapses and glutamate signaling. Korshunov *et al.*²⁴ identified TCL1 and TCL2 transcriptional subtypes of SHH MBs, where TCL2 tumors have high expression of neuronal genes and come from patients with exceptional survival rates.

In addition to these three bulk omics experiments, we also included cell types from three scRNA-seq studies. Riemondy *et al.* (2022)⁸ analyzed SHH MBs and found six clusters of tumor cells. SHH-A cells are associated with proliferation (SHH-A1 with S phase and SHH-A2 with G2M phase), while SHH-B cells are progenitors, with SHH-B1 resembling GCPs and SHH-B2 having high ribosomal expression. Lastly, SHH-C1 cells are enriched for RNA processing and axo-dendritic transport genes, while SHH-C2 cells have high levels of neural development markers like STMN2. We also generated signatures for cell types found during cerebellar GN development and considered studies from both human⁴⁴ and mouse⁶.

- 2. A figure, similar to figure 1C, reporting SHH-MB subtypes associated with distinct granule development stages, specific biomarkers, CNVs, and metabolites, should be included in the main text.**

This is a good suggestion and we have added a new Figure (Figure 7, page 21) at the end as a summary for the different proteins and molecular subtypes that are associated with each developmental region.

- 3. Given the frequency of CNVs in SHH-MB, has the association with other chromosomal alterations found in childhood SHH-MB (i.e. 17p-, 2+) been investigated?**

We appreciate this comment and hope that our revision will make this analysis clearer. Even though we only showed the results for a select few CNVs, we performed the analysis on all available chromosomal CNVs. We focused on 9q and 10q in the manuscripts because they were the strongest trends and were consistently significant no matter the gene set size considered. For 17p loss, we noticed that is enriched in tumors that have higher activation of cycling progenitors compared to non-cycling progenitors and we found the 2p gain is associated with higher scores for ribosomal progenitors.

To address these comments explicitly, we created a new supplementary table (Supplementary Table 4) that shows the Mann-Whitney p-values for every comparison, so the readers can analyze any pattern that interests them.

References:

1. Darmanis S, Sloan SA, Zhang Y, Enge M, Caneda C, Shuer LM, et al. A survey of human brain transcriptome diversity at the single cell level. *Proc Natl Acad Sci U S A*. 2015;
2. Cuevas-Diaz Duran R, González-Orozco JC, Velasco I, Wu JQ. Single-cell and single-nuclei RNA sequencing as powerful tools to decipher cellular heterogeneity and dysregulation in neurodegenerative diseases. *Front. Cell Dev. Biol.* 2022.
3. Zanin JP, Abercrombie E, Friedman WJ. Proneurotrophin-3 promotes cell cycle withdrawal of developing cerebellar granule cell progenitors via the p75 neurotrophin receptor. *Elife*. 2016;
4. Darnell JC, Van Driesche SJ, Zhang C, Hung KYS, Mele A, Fraser CE, et al. FMRP stalls ribosomal translocation on mRNAs linked to synaptic function and autism. *Cell*. 2011;
5. Sturman JA, Moretz RC, French JH, Wisniewski HM. Taurine deficiency in the developing cat: Persistence of the cerebellar external granule cell layer. *J Neurosci Res*. 1985;
6. Maar T, Morán J, Schousboe A, Pasantes-Morales H. Taurine deficiency in dissociated mouse cerebellar cultures affects neuronal migration. *Int J Dev Neurosci*. 1995;
7. Wang L, Xing X, Zeng X, Jackson SRE, TeSlaa T, Al-Dalahmah O, et al. Spatially resolved isotope tracing reveals tissue metabolic activity. *Nat Methods*. 2022;
8. Bemis KD, Harry A, Eberlin LS, Ferreira C, Van De Ven SM, Mallick P, et al. Cardinal: An R package for statistical analysis of mass spectrometry-based imaging experiments. *Bioinformatics*. 2015;

Reviewers' Comments:

Reviewer #1:

Remarks to the Author:

The authors have adequately addressed all of my comments. The study will be of high interest to the field.

Reviewer #2:

Remarks to the Author:

The authors presented RNA-seq and multiplexed imaging to support their findings of the heterogeneity observed in Medulloblastoma (MB) tumors with sonic hedgehog (SHH) activation, possibly due to differences in developmental pathway.

The reviewer's comments have largely been addressed. Remaining concerns:

1) Please replace the multiplexed spatial omics assays with multiplexed proteomic imaging and MALDI imaging because the presented results are too few markers and the rich data aspect of spatial omics has not been demonstrated. It is misleading to mention that "spatial omics" have been implemented. This is because the analysis of CyCIF data presented is premature (just one marker Ki67) and multiplexing is only two cycles.

2) Figure quality is still an issue. All the font sizes in the manuscript should be enlarged and white space in the figure should be minimized. For instance, UMAP axis labels are too small to read.

3) To reproduce figures, all the analysis and data should be provided in this link provided with the manuscript: <https://github.com/fraenkel-lab/shh-mben>

Currently, it is a broken link that provides 404 error.

Reviewer #3:

Remarks to the Author:

The authors addressed the raised concerns. The manuscript is clearly suitable for publication.

Comment 1: "Please replace the multiplexed spatial omics assays with multiplexed proteomic imaging and MALDI imaging because the presented results are too few markers and the rich data aspect of spatial omics has not been demonstrated. It is misleading to mention that "spatial omics" have been implemented. This is because the analysis of CyCIF data presented is premature (just one marker Ki67) and multiplexing is only two cycles"

We changed "multiplexed spatial omics assays" to "using multiplexed proteomic imaging and MALDI Imaging Mass Spectrometry". Additionally, we changed references to CyCIF throughout the manuscript to "multiplexed immunohistochemistry".

Comment 2: "Figure quality is still an issue. All the font sizes in the manuscript should be enlarged and white space in the figure should be minimized. For instance, UMAP axis labels are too small to read. "

Thank you for this comment. We have addressed this comment by making the following changes...

Figure 1: increasing font sizes for titles and axes on Figures 1B and 1D and the labels for Figure 1C. Additionally we moved the legend for Figure 1D to below the image, so that we could raise the font size.

Figure 2: We increased all font sizes for Figure 2A and for the legend and axes on Figure 2B.

Figure 3: We increased the font size for all values and axes in 3A. For 3B, we reduced the size of the figure and increased the font for the axes and titles. For 3C and 3D, we increased the font size for the axes and made the x axis labels more readable.

Figure 4: For 4A, we increased the font size for the title, axes, and legend. For 4B, we made the title font larger. For each sub-image in figure 4C, we decreased the image size and increased the font size for the titles and image axes. For 4D, we made the font larger for the title and axes.

Figure 5: We increased the size of the legend font for figures 5A, 5B, and 5C. We also increased the font size for all titles in 5A and 5D.

Figure 6: We made the font sizes larger for the text in figures 6A, 6C, and 6E. For 6B and 6D, we made the legends and the axes larger.

Figure 7: We increased the font size for all titles in the figure.

Comment 3: "To reproduce figures, all the analysis and data should be provided in this link provided with the manuscript: <https://github.com/fraenkel-lab/shh-mben>. Currently, it is a broken link that provides 404 error. "

We apologize for this. The link should now work, and it includes the code necessary for performing all analyses and figure generation. Additionally, there is a file titled "file_info.txt" that describes the contents of each individual code file and gives links to download the relevant data.

Reviewers' Comments:

Reviewer #2:

Remarks to the Author:

The authors have addressed the concerns about 1. figure quality, 2. GitHub, and 3. multiplexed data. I would recommend publication.